# Defining the chromatin-associated protein landscapes on *Trypanosoma brucei* repetitive elements using synthetic TALE proteins

Roberta Carloni[1,2†], Tadhg Devlin[1,2†‡], Pin Tong[1], Christos Spanos[1], Tanya Auchynnikava[1§], Juri Rappsilber[1,3], Keith R Matthews[2]*, Robin C Allshire[1]*

[1]Centre for Cell Biology and Institute of Cell Biology, School of Biological Sciences, University of Edinburgh, Edinburgh, United Kingdom; [2]Institute of Immunology and Infection Research, School of Biological Sciences, University of Edinburgh, Edinburgh, United Kingdom; [3]Institute of Biotechnology, Technische Universität, Berlin, Germany

*For correspondence:
keith.matthews@ed.ac.uk (KRM);
robin.allshire@ed.ac.uk (RCA)

†These authors contributed equally to this work

Present address: ‡NIBRT, Foster Avenue, Blackrock, Co. Dublin, Dublin, Ireland; §Proteomics Science Technology Platform, The Francis Crick Institute, London, United Kingdom

## eLife Assessment

This work significantly advances our understanding of chromatin organization within regions of repetitive sequences in the parasitic protozoan *Trypanosoma brucei*. Using cutting edge interdisciplinary tools, the authors provide **compelling** evidence for two discrete types of repetitive DNA element-associated proteins- one set involved in essential centromere function; and, the other involved in glycoprotein antigenic variation via homologous recombination. Thus, these **fundamental** findings have implications for this parasite's biology, and for therapeutic targeting in kinetoplastid diseases. This work will be exciting to those in the centromere/mitosis and parasite immunity fields.

[Editors' note: this paper was reviewed by Review Commons.]

**Abstract** Kinetoplastids, such as *Trypanosoma brucei*, are eukaryotes that likely separated from the main lineage at an exceptionally early point in evolution. Consequently, many aspects of kinetoplastid biology differ significantly from other eukaryotic model systems, including yeasts, plants, worms, flies, and mammals. As in many eukaryotes, the *T. brucei* genome contains repetitive elements at various chromosomal locations, including centromere- and telomere-associated repeats and interspersed retrotransposon elements. *T. brucei* also contains intermediate-sized and mini-chromosomes that harbour abundant 177 bp repeat arrays and 70 bp repeat elements implicated in Variable Surface Glycoprotein (VSG) gene switching. In many eukaryotes, repetitive elements are assembled in specialised chromatin such as heterochromatin; however, apart from centromere- and telomere-associated repeats, little is known about chromatin-associated proteins that decorate these and other repetitive elements in kinetoplastids. Here, we utilise affinity selection of synthetic TALE DNA binding proteins designed to target specific repeat elements to identify enriched proteins by proteomics. Validating the approach, a telomere repeat binding TelR-TALE identifies many proteins previously implicated in telomere function. Furthermore, the 70R-TALE designed to bind 70 bp repeats indicates that proteins involved in DNA repair are enriched on these elements that reside adjacent to VSG genes. Interestingly, the 177 bp repeat binding 177R-TALE enriches for many kinetochore proteins, suggesting that intermediate-sized and mini-chromosomes assemble kinetochores related in composition to those located on the main megabase chromosomes. This provides a first insight into the chromatin landscape of repetitive regions of the trypanosome

genome with relevance for their mechanisms of chromosome integrity, immune evasion, and cell replication.

## Introduction

Repetitive sequences are scattered across the genomes of many eukaryotes, where they define various functional chromosomal elements (*Bringaud et al., 2009*; *Feschotte, 2008*; *Kazazian, 2004*; *Slotkin and Martienssen, 2007*). For example, telomeres are generally composed of TG-rich repeats, added by the reverse transcriptase activity of telomerase, which uses its associated RNA as a template (*Li, 2021*; *Pfeiffer and Lingner, 2013*), whereas centromere regions often contain extensive tandem arrays of non-conserved repetitive sequences (*Allshire and Karpen, 2008*; *Miga and Alexandrov, 2021*; *Sullivan and Sullivan, 2020*; *Thakur et al., 2021*). In many eukaryotes, such arrays frequently provide a substrate for constitutive heterochromatin formation through di/tri-methylation of lysine 9 on histone H3 on resident nucleosomes. In addition, repetitive centromeric repeat arrays are associated with the assembly of specialised nucleosomes containing the centromere-specific histone H3 variant, generally known as CENP-A or cenH3 (*Allshire and Karpen, 2008*; *Talbert and Henikoff, 2020*). CENP-A nucleosomes form the foundation for kinetochore assembly, which mediates accurate chromosome segregation (*Allshire and Karpen, 2008*). Other repetitive sequences, such as transposable elements or their remnants, can alter – or have been co-opted to regulate – the expression of nearby genes (*Bourque et al., 2018*; *Fueyo et al., 2022*). In many eukaryotes, heterochromatin forms clusters in the nucleus that are generally located at the nuclear periphery or adjacent to nucleoli (*Bizhanova and Kaufman, 2021*; *van Steensel and Belmont, 2017*).

Kinetoplastids represent a distinct branch of protozoan eukaryotes within the Euglenozoa that diverged from the main eukaryotic lineage early during their evolution (*Cavalier-Smith, 2010*). As a result, kinetoplastids are distinct from most other eukaryotes in which cellular mechanisms are intensively studied, including yeasts, fungi, plants, nematodes, insects, and mammals. Many kinetoplastids are parasites that cause diseases in humans and economically important livestock. *Trypanosoma brucei*, for example, is prevalent in sub-Saharan Africa, where it is transmitted by tsetse flies and causes human African trypanosomiasis and Nagana in cattle (*Morrison et al., 2023*). Other kinetoplastid parasites that cause human diseases in the tropics include *Trypanosoma cruzi* (Chagas disease) and *Leishmania* spp. (leishmaniasis) (*Stuart et al., 2008*). Despite their divergence from most eukaryotes, the genomes of kinetoplastids contain a variety of repetitive sequences. The diploid genome of the commonly used laboratory *T. brucei* Lister 427 strain has recently been re-characterised with advanced genome assembly methods. The genome contains two homologues for each of the 11 large chromosomes, ranging in size from 900 to 4600 kb, 5–6 intermediate chromosomes, and ~100 mini-chromosomes (*Cosentino et al., 2021*; *Müller et al., 2018*; *Rabuffo et al., 2024*).

All *T. brucei* chromosomes are linear, and each end terminates with arrays of telomeric (TTAGGG)$_n$ repeats that are added by telomerase (*Sandhu and Li, 2017*). *T. brucei* exhibits the generally well-defined process of Variable Surface Glycoprotein (VSG) gene switching, which allows a proportion of parasites to evade the host immune system at any given time (*Barcons-Simon et al., 2023*). Most of the 2634 detected VSG genes are not expressed and reside in arrays in sub-telomeric regions, with others residing on mini-chromosomes. Only one VSG gene is expressed at any time, and only from one of the estimated 15 telomere adjacent bloodstream expression sites (BES) (*Cosentino et al., 2021*). The non-expressed VSG genes provide a library of potential alternative VSGs, so that the parasite has almost limitless potential to vary its protective coat. Further variation in the expressed VSG protein repertoire can be generated by recombination events between VSG genes and VSG pseudogenes, which comprise approximately 80% of the overall gene repertoire (*Mugnier et al., 2015*; *Cosentino et al., 2021*). Non-expressed VSG genes are exchanged with VSG genes residing in expression sites using recombination-directed processes that act on or near 70 bp repeats residing upstream of the resident VSG gene at each BES (*Boothroyd et al., 2009*; *Thivolle et al., 2021*). Apart from telomeric (TTAGGG)$_n$ repeats at their ends and one or two VSG genes, mini-chromosomes are comprised of tandem arrays of 177 bp repeats, which are also present on the poorly characterised intermediate-sized chromosomes (*Ersfeld, 2011*; *Sloof et al., 1983*; *Figure 1A*). The function of these 177 bp repeats is unknown, but mini-chromosomes have been shown to be maintained with high stability through mitotic cell divisions, suggesting that a mechanism is in place to ensure their

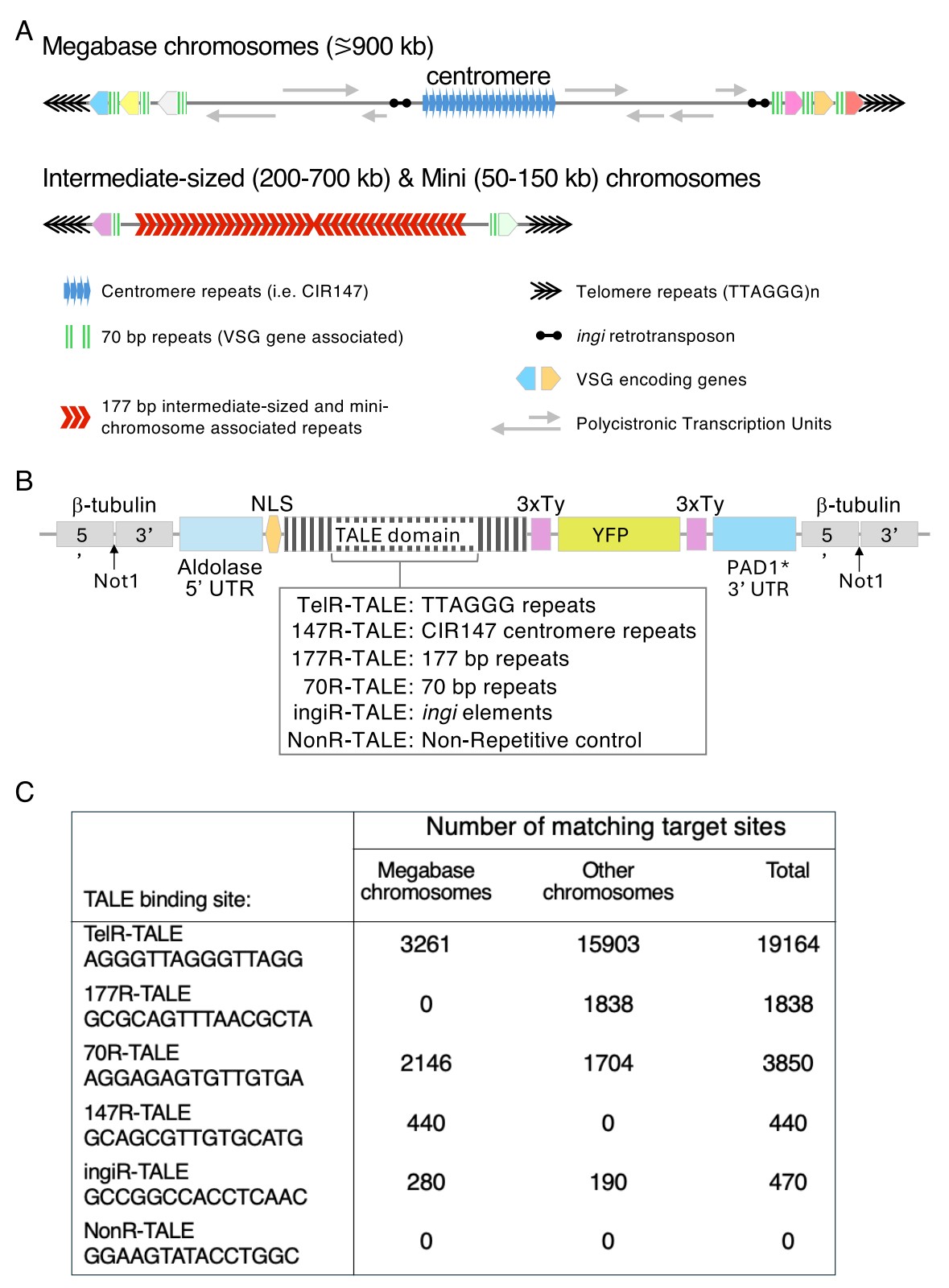

**Figure 1.** *T. brucei* repetitive elements, TALE design, and target site number. (**A**) Distinct repetitive elements are present at various locations on *T. brucei* chromosomes. (**B**) Construct designed to express the indicated TALE proteins that bind 15 bp target sequences fused to 3xTy1 and YFP tags when integrated at the β-tubulin locus. The Aldolase 5'UTR and PAD1 3'UTR regulate expression levels. A Bleomycin resistance marker gene provides Phleomycin selection (not shown). (**C**) Predicted number of target sequences for each TALE in the Lister 427 genome.

*Figure 1 continued*

The online version of this article includes the following source data and figure supplement(s) for figure 1:

**Figure supplement 1.** TALE-YFP construction and sequencing reveals rearranged TALE domain in TelR-TALE-YFP following integration in the *T. brucei* genome.

**Figure supplement 1—source data 1.** Original DNA-stained agarose gel for *Figure 1—figure supplement 1D* indicating the relevant PCR bands from *T. brucei* cells containing the indicated synthetic TALE-YFP protein expression constructs.

**Figure supplement 2.** Synthetic TALE proteins are expressed in Lister 427 *T. brucei* bloodstream-form cells, but the TelR-TALE protein is shorter than expected.

**Figure supplement 2—source data 1.** Original anti-YFP western for *Figure 1—figure supplement 2A* (left), indicating the relevant bands in *T. brucei* cells expressing the indicated synthetic TALE-YFP proteins.

**Figure supplement 2—source data 2.** Original anti-YFP western for *Figure 1—figure supplement 2A* (left), indicating the relevant bands in *T. brucei* cells expressing the indicated synthetic TALE-YFP proteins.

**Figure supplement 2—source data 3.** Original anti-Ty western for *Figure 1—figure supplement 2B*, indicating the relevant bands in *T. brucei* cells expressing the indicated synthetic TALE-YFP proteins.

**Figure supplement 3.** Growth assays of cells expressing TelR-TALE-GFP, 177R-TALE-GFP, or ingi-TALE-GFP, and their cellular localisation.

segregation with fidelity to daughter cells (*Ersfeld and Gull, 1997*; *Wickstead et al., 2003*). The main 11 megabase-sized chromosomes have been shown to assemble evolutionarily unconventional kinetochores composed of 25 kinetoplastid kinetochore proteins (KKT1-25) that mediate their accurate mitotic segregation and are distinctly different from those of other eukaryotes (*Akiyoshi and Gull, 2014*; *D'Archivio and Wickstead, 2017*; *Nerusheva et al., 2019*; *Akiyoshi and Gull, 2014*). ChIP-seq has shown that, on the main megabase-sized chromosomes, kinetochores assemble on different DNA sequences; on some chromosomes, kinetochores coincide with tandem arrays of CIR147 repeats or related repeat elements (*Akiyoshi and Gull, 2014*; *Echeverry et al., 2012*; *Obado et al., 2007*). CIR147 repeats produce non-coding transcripts that are processed by Dicer into siRNAs and loaded into Argonaute/TbAGO1 (*Tschudi et al., 2012*). In addition, SLAC and *ingi*-related retrotransposons are dispersed across the *T. brucei* genome (*Bringaud et al., 2009*) and are also transcribed and processed into Ago1-associated siRNA (*Tschudi et al., 2012*).

The key hallmarks of eukaryotic heterochromatin, di/tri-methylation of histone H3 on lysine 9 (H3K9) or lysine 27 (H3K27), cannot be detected in *T. brucei* or other kinetoplastids because their histones, including H3, are particularly divergent rendering useless most existing antibody reagents used for histone post-translational modification (PTM) analyses in other eukaryotes (*Deák et al., 2023*; *Figueiredo et al., 2009*). Thus, it is not known which, if any, other modified or unmodified residues on *T. brucei* histones might nucleate repressive chromatin that could be regarded as heterochromatin. However, mass spectrometry has identified a plethora of residues in *T. brucei* and *T. cruzi* histones that exhibit various PTMs (*de Lima et al., 2020*; *Kraus et al., 2020*; *Maree et al., 2022*; *Picchi et al., 2017*). Some of these PTMs may be involved in forming distinct chromatin structures on repetitive elements through the recruitment of specific proteins analogous to chromodomain protein recruitment via H3K9 or H3K27 methylation in other eukaryotes (*Allshire and Madhani, 2018*).

To characterise the chromatin context and the possible function of *T. brucei* repetitive elements, we applied an unbiased proteomics-based approach. We exploited synthetic DNA binding TALE (transcription activator-like effectors) fusion protein expression in *T. brucei* to bind to particular repetitive sequences and, following affinity selection, identify specific factors enriched on these chromosomal regions. Thus, synthetic TALE proteins were designed that were expected to bind the terminal telomeric $(TTAGGG)_n$ repeat arrays (TelR-TALE), the most frequent canonical CIR147 centromeric repeat (147R-TALE), core 177 bp repeats (177R-TALE), 70 bp BES-associated repeats (70R-TALE), *ingi*-related retrotransposon repeats (ingiR-TALE), and a Non-Recognised control (NonR-TALE) (*Figure 1B*). These synthetic TALE proteins were expressed as YFP fusion proteins with a nuclear localisation signal in *T. brucei* Lister 427 bloodstream-form cells with ChIP-seq confirming that they target the repeat elements that they were designed to bind. Validating the approach, affinity purification of TelR-TALE followed by proteomics analyses identified many proteins that were also enriched by affinity purification of the endogenous YFP-tagged *T. brucei* TRF telomere repeat binding protein. Further, several proteins involved in DNA-repair recombination were enriched with affinity-purified 70R-TALE, suggesting candidates that may be involved in mediating VSG gene switching events via these

repeats. Surprisingly, many kinetochore proteins were detected as being enriched on 177 bp repeats. Thus, intermediate-sized and mini-chromosomes may assemble kinetochores and utilise machinery related to that operating on the main 11 chromosomes for their accurate mitotic segregation.

## Results

### Synthetic TALE-YFP fusion proteins that target *T. brucei* repetitive sequences

Five synthetic transcription activator-like effector TALE proteins were designed that were predicted to specifically bind 15 bp target sequences residing in different repetitive elements using pre-assembled tetramer and trimer modules (*Moore et al., 2014*; *Figure 1B and C*; *Figure 1—figure supplement 1*). BLAST searches confirmed that each selected 15 bp target sequence was unique to the specific target repetitive element with no exact match elsewhere in the *T. brucei* 427 reference genome (*Cosentino et al., 2021*; *Rabuffo et al., 2024*). The five TALEs assembled were thus predicted to bind: (i) telomeric (TTAGGG)$_n$ repeats residing at all chromosome ends (TelR-TALE) (*Blackburn and Challoner, 1984*; *Van der Ploeg et al., 1984*), (ii) the 70 bp repeat arrays that reside upstream of bloodstream VSG gene expression sites, and in shorter tracts adjacent to silent subtelomeric VSG genes and contribute to VSG gene switching events (70R-TALE) (*Boothroyd et al., 2009*; *Glover et al., 2013*; *Hovel-Miner et al., 2016*; *Kim and Cross, 2010*; *Thivolle et al., 2021*), (iii) the satellite-like centromere-associated 147 bp Chromosome Internal Repeats (147R-TALE) (*Akiyoshi and Gull, 2014*; *Obado et al., 2005*; *Tschudi et al., 2012*), (iv) the 177 bp satellite repeats that are concentrated on mini- and intermediate-sized chromosomes (177R-TALE) (*Wickstead et al., 2004*), and (v) a sequence common to the *ingi* clade of non-LTR retrotransposon interspersed repeat elements (ingiR-TALE) (*Bringaud et al., 2008*). A control NonR-TALE protein was also designed, which was predicted to have no target sequence in the *T. brucei* genome. Each synthetic TALE DNA binding domain open reading frame (ORF) was fused at its N-terminus to DNA encoding the *T. brucei* La protein nuclear localisation signal (NLS) and at its C-terminus with DNA encoding a 3xTy-YFP tag (*Dean et al., 2015*; *Marchetti et al., 2000*). *T. brucei* genes are polycistronic with their expression regulated by RNA processing and turnover. Consequently, the attenuated D1-354 PAD1 3'UTR from the PAD1 gene was placed downstream of each NLS-TALE-3xTy-YFP ORF. Use of this 3'UTR, which drives high-level expression in the stumpy transmission stage of parasites but only low-level expression in proliferative bloodstream forms (*MacGregor and Matthews, 2012*) restricted TALE protein expression levels. All constructs carried the Aldolase (ALD) 5' UTR (ALD) to enable 5' end RNA processing. Each of the final ALD5'UTR-NLS-TALE-3xTy-YFP*PAD1-3'UTR plasmids was integrated by homologous recombination at the β-tubulin gene locus in monomorphic Lister 427 bloodstream-form *T. brucei* cells (for brevity hereon the constructs and proteins produced are referred to as ---R-TALEs; *Figures 1B–C*, *Figure 1—figure supplement 1A–C*, *Figure 1—figure supplement 2*, *Figure 1—figure supplement 3*).

Proteins extracted from resultant TelR-TALE, 70R-TALE, 147R-TALE, 177R-TALE, ingiR-TALE, and NonR-TALE *T. brucei* transformants were analysed by anti-GFP and anti-Ty westerns (*Figure 1—figure supplement 2*). Cell lines expressing representative TALE-YFP proteins displayed no fitness deficit (*Figure 1—figure supplement 3A*). Five of the six synthetic ORFs produced proteins of the expected size of ~110 kDa. However, the expression level of NonR-TALE-YFP was lower than other TALE-YFP proteins; this may relate to the lack of DNA binding sites for NonR-TALE-YFP in the nucleus. Moreover, the TelR-TALE protein was smaller than expected; further investigation revealed that the repetitive nature of the telomeric target sequence AGGGTTAGGGTTAGG gave rise to a 612 bp direct repeat within the TALE encoding modules which, following transformation of *T. brucei*, resulted in a deletion event that reduced the predicted recognised target sequence to 8 rather than 15 bases of telomeric repeat (*Figure 1—figure supplement 1D and E*). Nevertheless, about 19,000 copies of the (TTAGGG)$_n$ sequence reside at *T. brucei* telomeres and contain the predicted, albeit truncated, TelR-TALE target sequence AGGGTTAG. Indeed, further analysis confirmed that the TelR-TALE-YFP protein binds telomeres in vivo (see below).

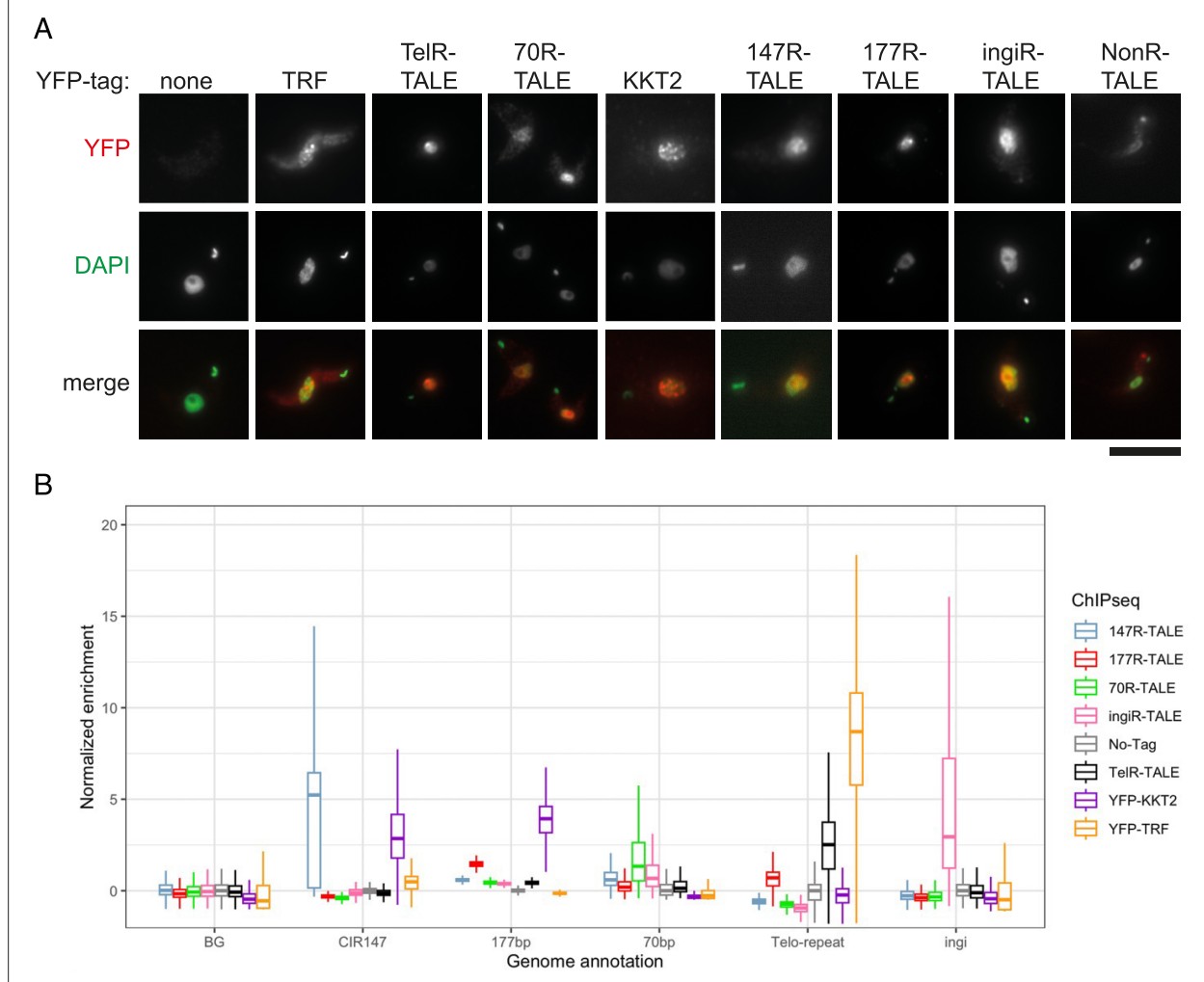

**Figure 2.** Localisation and specific target sequence association of five synthetic TALE-YFP fusion proteins expressed in *T. brucei* compared to YFP-TRF and YFP-KKT. (**A**) Bloodstream-form Lister 427 *T. brucei* cells expressing the indicated TALE-YFP fusion proteins fixed and TALE-YFP protein localisation detected with anti-GFP primary antibody and Alexa Fluor 568-labelled secondary antibody (red). Nuclear and kinetoplast (mitochondrial) DNA were stained with DAPI (green). Control cells expressing telomeric YFP-TRF, centromeric YFP-KKT2 kinetochore protein, or wild-type Lister 427 cells expressing no YFP are also shown. Scale bar, 10 μm. (**B**) Anti-GFP ChIP-seq analysis for 147R-TALE, 177R-TALE, 70R-TALE, TelR-TALE, and ingiR-TALE, demonstrating that each protein is enriched on the repeat elements they were designed to recognise: CIR147 repeats, 177 bp repeats, 70 bp repeats, telomeric (TTAGGG)$_n$ repeats and ingi retrotransposons. Enrichments obtained for the YFP-KKT2 kinetochore protein, the TRF telomere repeat binding protein, and with a No-Tag control are shown for comparison. Data are from two biological replicates.

The online version of this article includes the following figure supplement(s) for figure 2:

**Figure supplement 1.** Fields of *T. brucei* cells showing the cellular localisation of six expressed synthetic TALE-YFP fusion proteins compared to YFP-TRF and YFP-KKT.

## Synthetic repeat targeting TALE proteins localise to nuclei and are enriched on their cognate sequences

To determine the localisation of the six TALE proteins, anti-GFP immunolocalisation was performed on *T. brucei* cells expressing each individual TALE-YFP fusion protein or, as controls, the YFP-TRF telomere (TTAGGG)$_n$ binding protein or YFP-KKT2 centromere-associated kinetochore protein (*Figure 2*, *Figure 2—figure supplement 1*, *Figure 1—figure supplement 3B*). All synthetic TALE-YFP proteins and the endogenously tagged YFP-TRF and YFP-KKT2 proteins localised within nuclei, with YFP-TRF and YFP-KKT2 exhibiting distinct nuclear foci as expected for telomeres and centromeres (*Li, 2023*; *Akiyoshi and Gull, 2014*). The TelR-TALE-YFP and 147R-TALE-YFP localisation patterns were also punctate and comparable to that of YFP-TRF and YFP-KTT2, respectively. Furthermore, the

localisation pattern for 177R-TALE-YFP was consistent with the known location of mini-chromosome 177 bp repeats around the nuclear periphery (*Ersfeld and Gull, 1997*). Both the 70R-TALE-YFP and ingiR-TALE-YFP proteins exhibited a diffuse nuclear signal with no specific sub-nuclear pattern. NonR-TALE-YFP displayed a diffuse nuclear and cytoplasmic signal; unexpectedly, the cytoplasmic signal appeared to be in the vicinity of the kDNA of the kinetoplast (mitochondria). We note that artefactual localisation of some proteins fused to an eGFP tag has previously been observed in *T. brucei* (*Pyrih et al., 2023*).

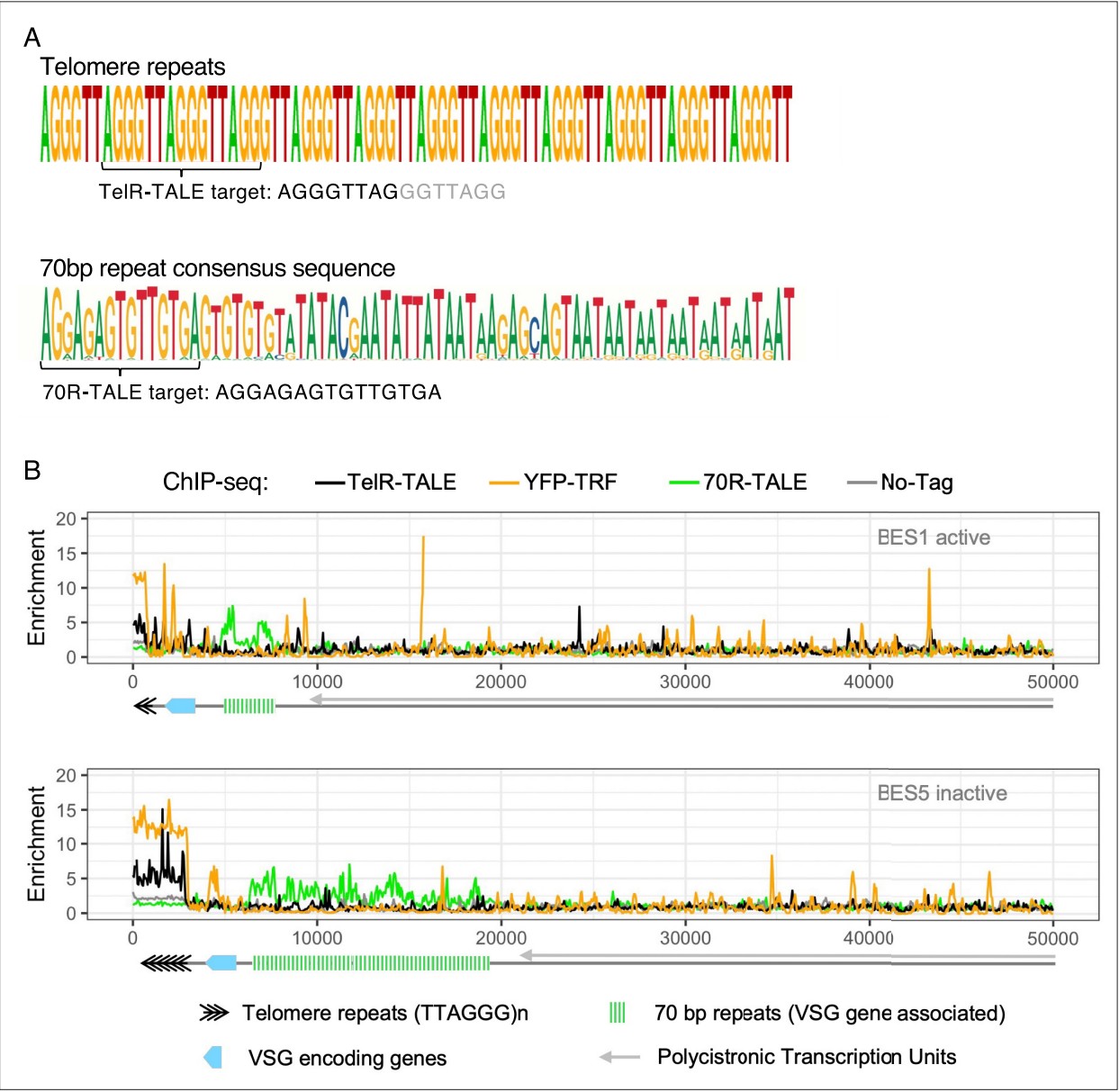

**Figure 3.** TelR-TALE-YFP and 70R-TALE-YFP are enriched at or near telomeric *T. brucei* bloodstream expression sites. (**A**) Telomeric repeat (TTAGGG)$_n$ sequence (top) and 70 bp repeat consensus sequence (bottom). Sequences that TelR-TALE and 70R-TALE were designed to bind are indicated. Deletion of TelR-TALE recognition modules following integration in *T. brucei* results in recognition of AGGGTTAG rather than the full 15 bp target sequence. (**B**) Anti-GFP ChIP-seq for cells expressing TelR-TALE-YFP, YFP-TRF, or 70R-TALE–YFP proteins, or 427 cells expressing no YFP-tagged protein. Anti-GFP ChIP-seq enrichment profiles are shown for telomeric bloodstream expression sites (BES) BES1 (top) and BES5 (bottom). Diagrams show the position of telomeric (TTAGGG)$_n$ repeats (black chevrons), VSG genes (blue), and upstream 70 bp repeats (green bars). Data are from two biological replicates. Y axis: log$_2$ values, X axis: base pairs.

The online version of this article includes the following figure supplement(s) for figure 3:

**Figure supplement 1.** IngiR-TALE is enriched at matching binding sites located in retrotransposons.

To determine if the TALE proteins were enriched on the repetitive elements that they were designed to bind, anti-GFP ChIP-seq was performed. The resulting ChIP-seq reads were aligned to the most recent *T. brucei* 427 genome assembly (*Cosentino et al., 2021*; *Rabuffo et al., 2024*) and the relative specificity compared (*Figure 2B*). The truncated TelR-TALE protein predicted to bind AGGGTTAG within telomeric (TTAGGG)$_n$ arrays (*Figure 3A*) was found to be enriched at the end of all megabase-sized, intermediate-sized, and mini-chromosomes coincident with telomere repeat binding protein YFP-TRF enrichment (e.g. *Figure 3B*). The 70R-TALE bound to 70 bp repeats (*Figure 3A*) that reside upstream of many VSG gene BES loci, regardless of their expression status, 2–8 kb from terminal (TTAG GG)$_n$ telomere repeat arrays (*Hertz-Fowler et al., 2008*) (binding at active BES1 and inactive BES5 is shown in *Figure 3B*). The ingiR-TALE protein was enriched over the 470 matching *ingi* element target sites dispersed across the *T. brucei* genome and, as expected, these included the region of similarity in RIME, SIDER, and DIRE retrotransposons (*Bringaud et al., 2008*; *Figure 3—figure supplement 1*).

The centromere region of *T. brucei* chromosomes 4, 5, and 8 contains extensive arrays of canonical CIR147 repeats. Divergent but related repeats are associated with the centromeres of the other main chromosomes, but no CIR147-related centromere repeats reside on the intermediate-sized or mini-chromosomes. Hence, 147R-TALE, which was designed to bind the TTGACGTGAAAATAC sequence within the consensus CIR147 repeat (*Figure 4A*), and for which homologous siRNAs are produced (*Patrick et al., 2009*; *Tschudi et al., 2012*), showed enrichment on the cognate repeat arrays at centromeres 4 and 5, and to some extent centromere 8, which are also occupied by the YFP-KKT2 kinetochore protein (*Figure 4B and C*). In contrast, the 147R-TALE did not decorate the CIR147-related repeats residing at centromeres 9, 10, and 11 or the more divergent classes of repeats bound by YFP-KKT2 at centromeres 1, 2, 3, 6, and 7 (*Figure 4C*). ChIP-seq analysis for the 177R-TALE showed that this synthetic protein was enriched on target intermediate-sized and mini-chromosome 177 bp repeat arrays (*Figure 5*).

## TelR-TALE affinity purification verifies the use of TALEs to identify repetitive element-associated proteins

All five synthetic TALE-YFP proteins were found to target the repetitive elements to which they were designed to bind when expressed in *T. brucei*. At least five proteins have previously been shown to be specifically enriched with the *T. brucei* TRF telomere binding protein in affinity purifications: TIF2, TelAP1, TelAP2, TelAP3, and PolQ/PolIE (*Leal et al., 2020*; *Reis et al., 2018*; *Weisert et al., 2024*). Therefore, to test if repeat-targeted TALE-YFP proteins could be used to identify proteins associated with repetitive elements, we affinity-purified solubilised TelR-TALE-bound chromatin and compared the associated proteins with those we detected as being enriched with YFP-TRF by mass spectrometry (AP-LC-MS/MS; *Figure 6A and B*; *Figure 6—figure supplement 1*; *Supplementary file 1a and b*). As expected, known telomere-associated proteins TRF (Tb927.10.12850), TIF2 (Tb927.3.1560), TelAP1 (Tb927.11.9870), TelAP2 (Tb927.6.4330), TelAP3 (Tb927.9.4000), RAP1 (Tb927.11.370), and PolQ/PolIE (Tb927.11.5550) were enriched with affinity-purified YFP-TRF (*Figure 6A*, *Figure 6—figure supplement 1*, *Supplementary file 1a*). In addition, replication/repair proteins RPA2 (Replication Factor A; Tb927.11.9130) and PPL2 (PrimPol-Like protein 2; Tb927.10.2520) were also enriched with YFP-TRF, along with the RNA binding proteins ZC3H39 (Tb927.10.14930) and ZC3H40 (Tb927.10.14950), HDAC3 (Tb927.2.2190), and histones (*Figure 6A*). Using the same affinity selection procedure, an overlapping set of 108 proteins was found to be enriched with TelR-TALE-YFP-bound chromatin (*Figure 6B*, *Figure 6—figure supplement 1*, *Supplementary file 1b*); these included TRF, TIF2, TelAP1, TELAP2, TELAP3, PolQ/PolIE, PPL2, HDAC3, and histones; however, RAP1 was only weakly enriched. In addition, all three Replication Factor A subunits (RPA1, 2, 3; Tb927.11.9130, Tb927.5.1700, Tb927.9.11940) were enriched with TelR-TALE, but only RPA2 with YFP-TRF. Notably, two RNA-associated proteins PABP2 (Tb927.9.10770) and MRB1590 (Tb927.3.1590), which were previously identified as potential telomere-associated proteins (*Reis et al., 2018*; *Weisert et al., 2024*), were detected in both our YFP-TRF and TelR-TALE affinity purifications. Moreover, the ZC3H39 (Tb927.10.14930) and ZC3H40 (Tb927.10.14950) RNA binding proteins, which heterodimerise to regulate respiratome transcript levels (*Trenaman et al., 2019*), were enriched in affinity selections of both proteins (*Figure 6—figure supplement 1*).

Overall, these data indicate that a core set of known telomere/TRF-associated proteins were also enriched with the synthetic TelR-TALE telomere binding protein. Thus, we conclude that our other

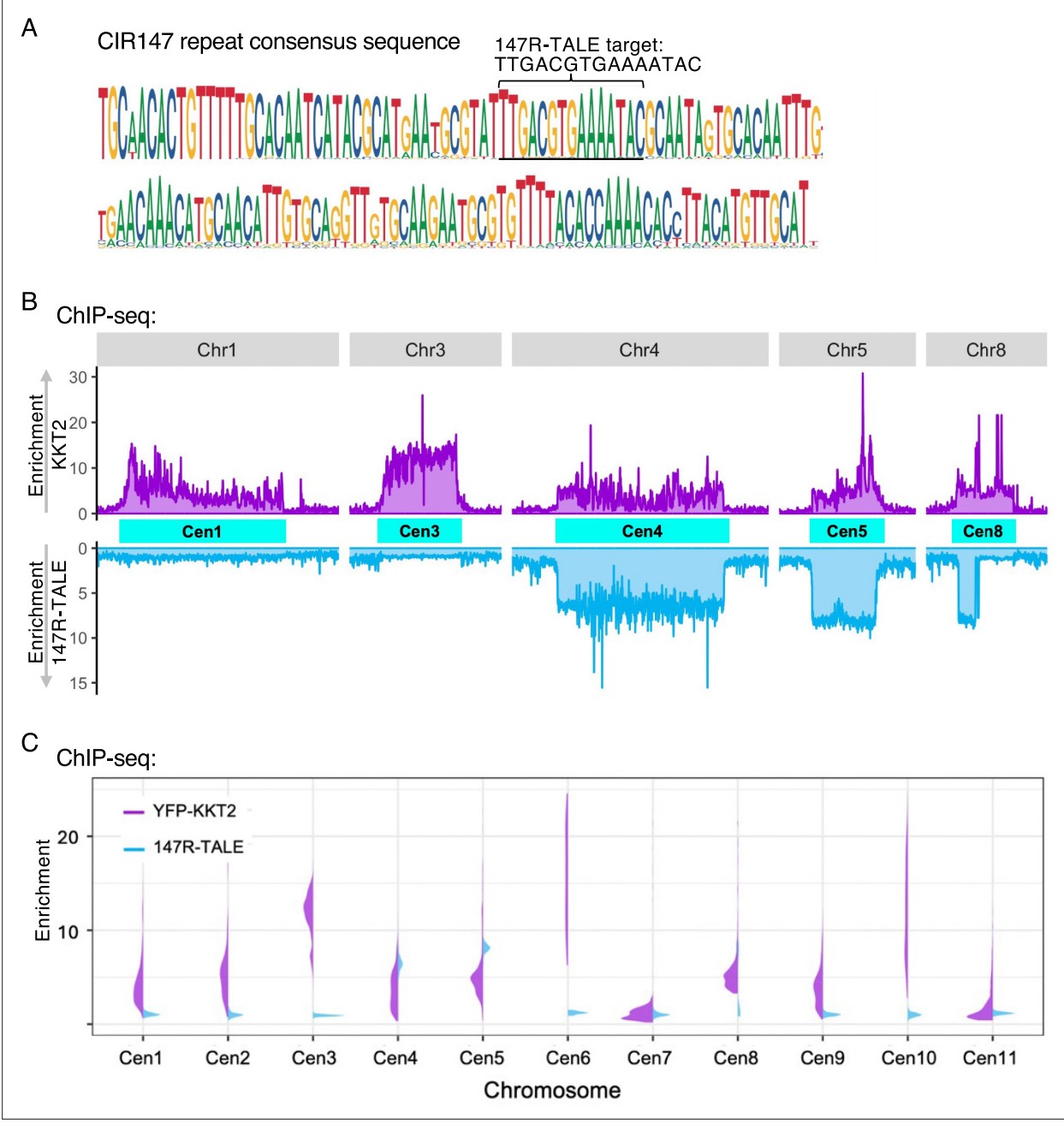

**Figure 4.** The 147R-TALE-YFP protein is enriched at a subset of centromeres containing canonical CIR147 repeats. (**A**) CIR147 repeat consensus sequence. Sequence that 147R-TALE-YFP was designed to bind is indicated. (**B**) Comparison of sequences enriched in YFP-KKT2 (purple) and 147R-TALE-YFP (blue) anti-GFP ChIP-seq for chromosomes 1, 3, 4, 5, and 8. DNA from all centromeres is enriched in YFP-KKT2 anti-GFP ChIP-seq, whereas only CIR147 repeats at centromeres on chromosomes 4, 5, and 8 are enriched in 147R-TALE-YFP anti-GFP ChIP-seq. (**C**) Split-Violin plot demonstrating the relative enrichment of YFP-KKT2 (purple) and 147R-TALE-YFP (blue) over the 11 main chromosome centromere regions. Data are from two biological replicates. Y axis: $\log_2$ values.

synthetic TALE-YFP proteins, designed to bind distinct repetitive elements, could allow the iden-tification of proteins specifically residing on those other sequences in vivo. Moreover, a similar set of enriched proteins was identified in TelR-TALE-YFP affinity purifications when compared with cells expressing no YFP fusion protein (No-YFP), the NonR-TALE-YFP, or the ingiR-TALE-YFP as controls (***Figure 6—figure supplement 2B***, ***Figure 6—figure supplement 3A***; ***Supplementary file 1c, d, and o***). Thus, the most enriched proteins are specific to TelR-TALE-YFP-associated chromatin rather than to the TALE-YFP synthetic protein module or other chromatin.

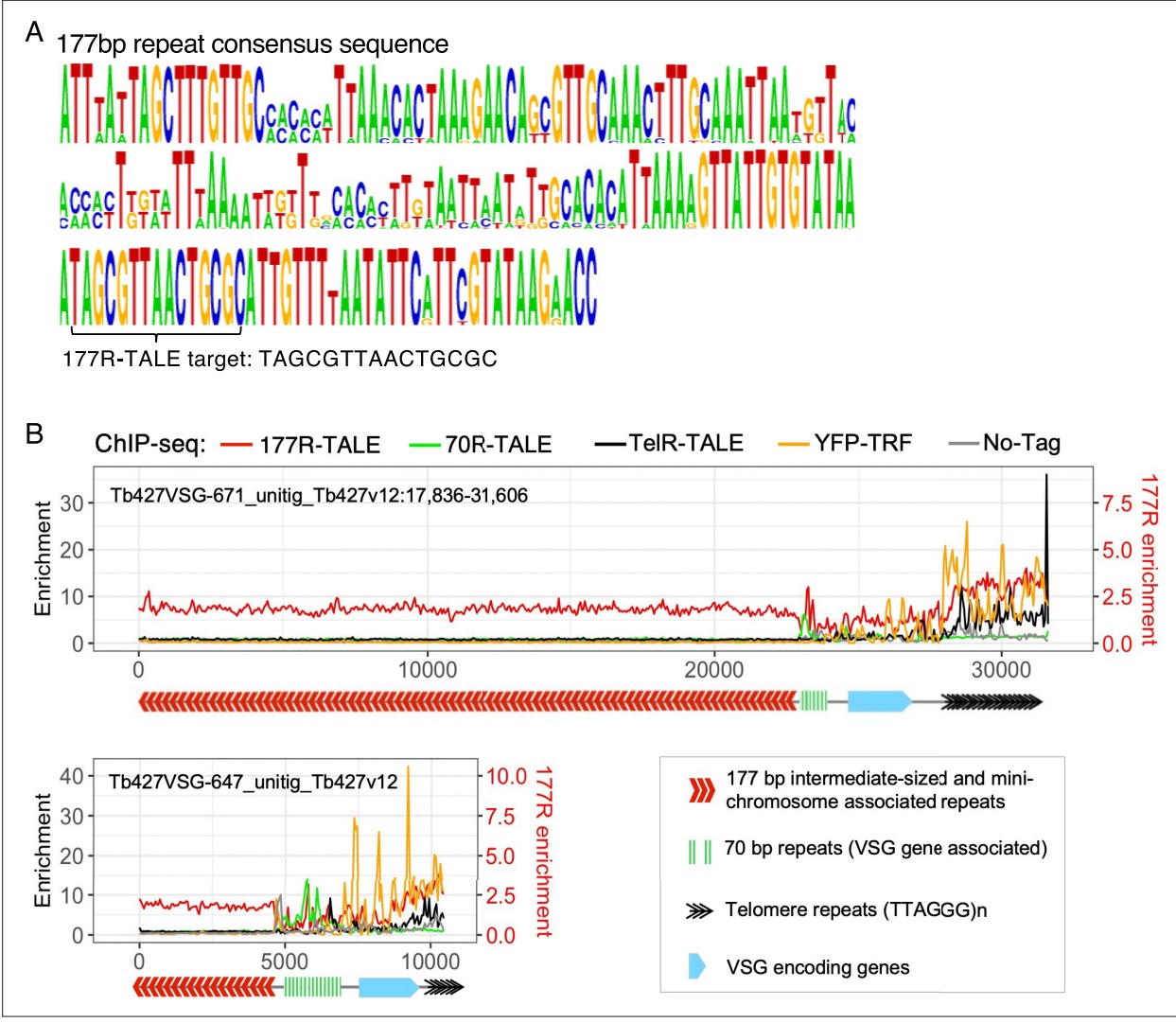

**Figure 5.** The 177R-TALE-YFP is enriched over 177 bp repeats located on intermediate-sized and mini-chromosomes. (**A**) 177 repeat consensus sequence. Sequence that 177R-TALE-YFP was designed to bind is indicated. (**B**) Distribution of 177R-TALE-YFP, TelR-TALE-YFP, YFP-TRF, and 70R-TALE-YFP, at two intermediate/mini-chromosome telomeres determined by anti-GFP ChIP-seq. Anti-GFP ChIP-seq of 427 cells expressing no tagged protein is included as control. Diagrams below ChIP-seq profiles indicate the positions of 177 bp repeats (red chevrons), 70 bp repeats (green bars), VSG encoding genes (blue), and telomere (TTAGGG)$_n$ repeats (black chevrons) within Tb427VSG-671_unitig_Tb427v12:17,836–31,606 (31kb) and Tb427VSG-647_unitig_Tb427v12 (10 kb). Data are from two biological replicates. Y axis: log$_2$ values, X axis: base pairs.

## Target sequence copy number may determine effectiveness of TALE-YFP proteins in identifying repeat-associated proteins

We estimated that the most recent *T. brucei* 427 genome assembly contains 19,164 copies of the telomeric AGGGTTAG target sequence which the truncated Tel-TALE-YFP is predicted to bind within (TTAGGG)$_n$ repeat arrays (*Cosentino et al., 2021*; *Rabuffo et al., 2024*). In contrast, there are only 440 and 470 targets matching the predicted binding sites TTGACGTGAAAATAC and GCCGGCACCTCAAC for the 147R-TALE and ingiR-TALE synthetic proteins, respectively (*Figure 1C*). NonR-TALE is predicted to have no matching binding sites in the *T. brucei* TREU 427 genome. To determine if proteins associated with such low copy number TALE-YFP target sequences could be identified, we applied the same AP-LC-MS/MS proteomics procedure to *T. brucei* cells expressing 147R-TALE, ingiR-TALE, or NonR-TALE. Comparison of either 147R-TALE or ingiR-TALE affinity purifications results with the No-YFP or NonR-TALE-YFP control affinity purifications showed no specific enrichment of any proteins of obvious potential functional interest with either 147R-TALE or ingiR-TALE (*Figure 6—figure supplement 2E and F*, *Figure 6—figure supplement 4*, *Supplementary file 1e–h*). Thus,

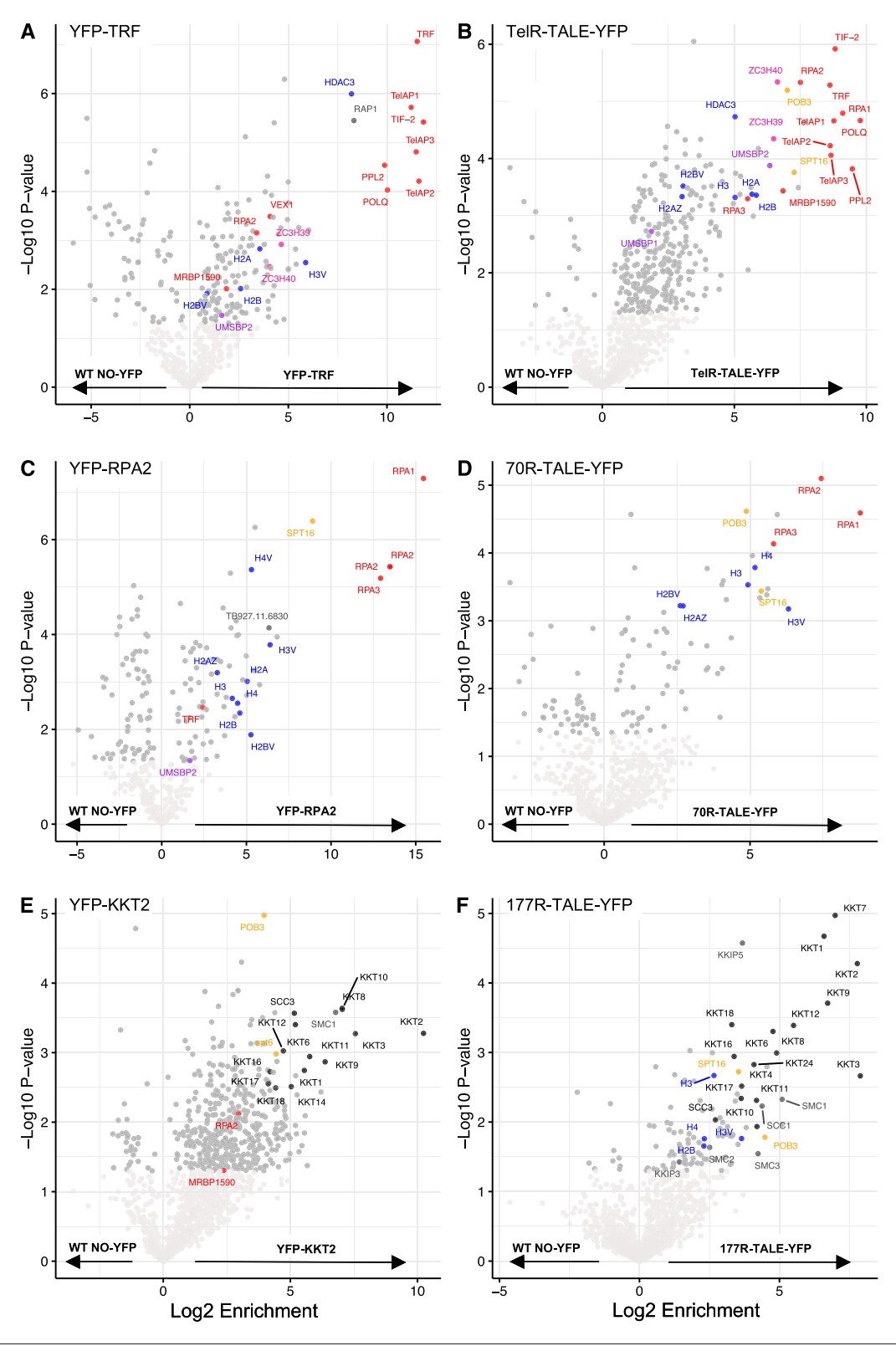

**Figure 6.** Affinity selection of TelR-TALE-YFP enriches for telomere-associated proteins and 177R-TALE-YFP protein enriches for kinetochore proteins. Affinity selection was performed on control cells expressing YFP-TRF (**A**), YFP-RPA2 (**C**), YFP-KKT2 (**E**), or No-YFP-tagged protein, and cells expressing synthetic TelR-TALE-YFP (**B**), 70R-TALE-YFP (**D**), 177R-TALE-YFP (**F**). Enriched proteins were identified and quantified by LC-MS/MS analysis

*Figure 6 continued on next page*

*Figure 6 continued*

relative to the No-YFP tag control. The data for each plot is derived from three biological replicates. Cut-offs used for significance: $p < 0.05$ (Student's *t*-test). Enrichment scores for proteins identified in each affinity selection are presented in *Supplementary file 1*.

The online version of this article includes the following figure supplement(s) for figure 6:

**Figure supplement 1.** Overlap of proteins enriched in affinity purifications of both synthetic protein telomere binding protein TelR-TALE-YFP and YFP-TRF.

**Figure supplement 2.** A control TALE that binds no specific *T. brucei* sequence validates proteins enriched in TelR-TALE, 70R-TALE, and 177R-TALE affinity purifications.

**Figure supplement 3.** Affinity selection of TelR-TALE-YFP, 70R-TALE-YFP 177R-TALE-YFP relative to ingiR-TALE-YFP validates specificity.

**Figure supplement 4.** No proteins of interest are detected following affinity selection of 147R-TALE or ingiR-TALE.

**Figure supplement 5.** Overlap of proteins enriched in affinity purifications of both kinetochore protein YFP-KKT2 and synthetic protein 177R-TALE.

---

the nuclear ingiR-TALE-YFP provides an additional chromatin-associated negative control for affinity purifications with the TelR-TALE-YFP, 70R-TALE-YFP, and 177R-TALE-YFP proteins (*Figure 6—figure supplement 3*, *Supplementary file 1o-q*). Moreover, although kinetochore proteins are enriched on CIR147 repeats (*Figure 4B and C*; *Akiyoshi and Gull, 2014*), no kinetochore proteins were detected in 147R-TALE affinity purifications. Thus, although ChIP-seq showed that both 147R-TALE and ingiR-TALE were enriched on their cognate target sequences, it appears that there are insufficient copies of these repeats for our AP-LC-MS/MS procedure to reveal associated proteins above background. We therefore focused our attention on the 70R-TALE and 177R-TALE synthetic proteins for which there are 3850 and 1828 predicted binding sites in the genome, respectively (*Figure 1C*).

## The RPA complex is enriched with synthetic 70 bp repeat binding protein

70R-TALE affinity purifications showed enrichment of all three subunits of the Replication Protein A complex (RPA1, RPA2, and RPA3) comparable to the enrichment detected in affinity purification of YFP-RPA2 itself (*Figure 6C and D*; *Supplementary file 1i and j*). Proteins identified as being enriched with 70R-TALE-YFP (*Figure 6D*) were similar in comparisons with either the No-YFP, NonR-TALE-YFP, or ingiR-TALE-YFP as negative controls (*Figure 6—figure supplements 1 and 2C*, *Figure 6—figure supplement 3B*; *Supplementary file 1k and p*). Along with the RPA complex, FACT subunits (SPT16 and POB3), histones, and Tb927.11.6830 were also enriched with both YFP-RPA2 and 70R-TALE-YFP affinity purification. This collection of proteins was also enriched in affinity purifications of TelR-TALE, which binds terminal telomeric (TTAGGG)$_n$ repeats. In contrast, the 70R-TALE targets 70 bp repeats residing several kilobase pairs internal from telomeres (ChIP-seq, *Figure 3B*). Given the known role for the RPA complex in DNA repair and replication, it may have distinct roles in mediating specific DNA transactions via 70 bp repeats and in telomere repeat dynamics (*Boothroyd et al., 2009*; *Li, 2023*).

## Kinetochore proteins are enriched on 177 bp repeats bound by 177R-TALE

In contrast to 70R-TALE and TelR-TALE, affinity selection of the 177R-TALE resulted in enrichment of a distinct set of proteins which unexpectedly included 14 of the 25 known kinetoplastid core kinetochore proteins: KKT1, KKT2, KTT3, KKT4, KKT6, KKT7, KKT8, KKT9, KKT10, KKT11, KKT12, KKT16, KKT17, KKT24 (*Akiyoshi and Gull, 2014*; *D'Archivio and Wickstead, 2017*; *Nerusheva et al., 2019*; *Figure 6F*; *Supplementary file 1l*). The same kinetochore proteins were enriched regardless of whether the 177R-TALE proteomics data was compared with No-YFP, NonR-TALE, or ingiR-TALE controls (*Figure 6—figure supplement 2D*, *Figure 6—figure supplement 3C*, *Supplementary file 1m and q*). For comparison, YFP-KKT2 was affinity-selected from *T. brucei* cells expressing endogenous N-terminal YFP-tagged KKT2 (*Figure 6E*; *Supplementary file 1n*). A clearly overlapping set of proteins was detected in both 177R-TALE-YFP and YFP-KKT2 affinity purifications (*Figure 6—figure supplement 5*). Moreover, the outer kinetochore-associated proteins KKIP3 and KKIP5, which transiently associate with *T. brucei* kinetochores through Aurora B kinase regulation, were also enriched

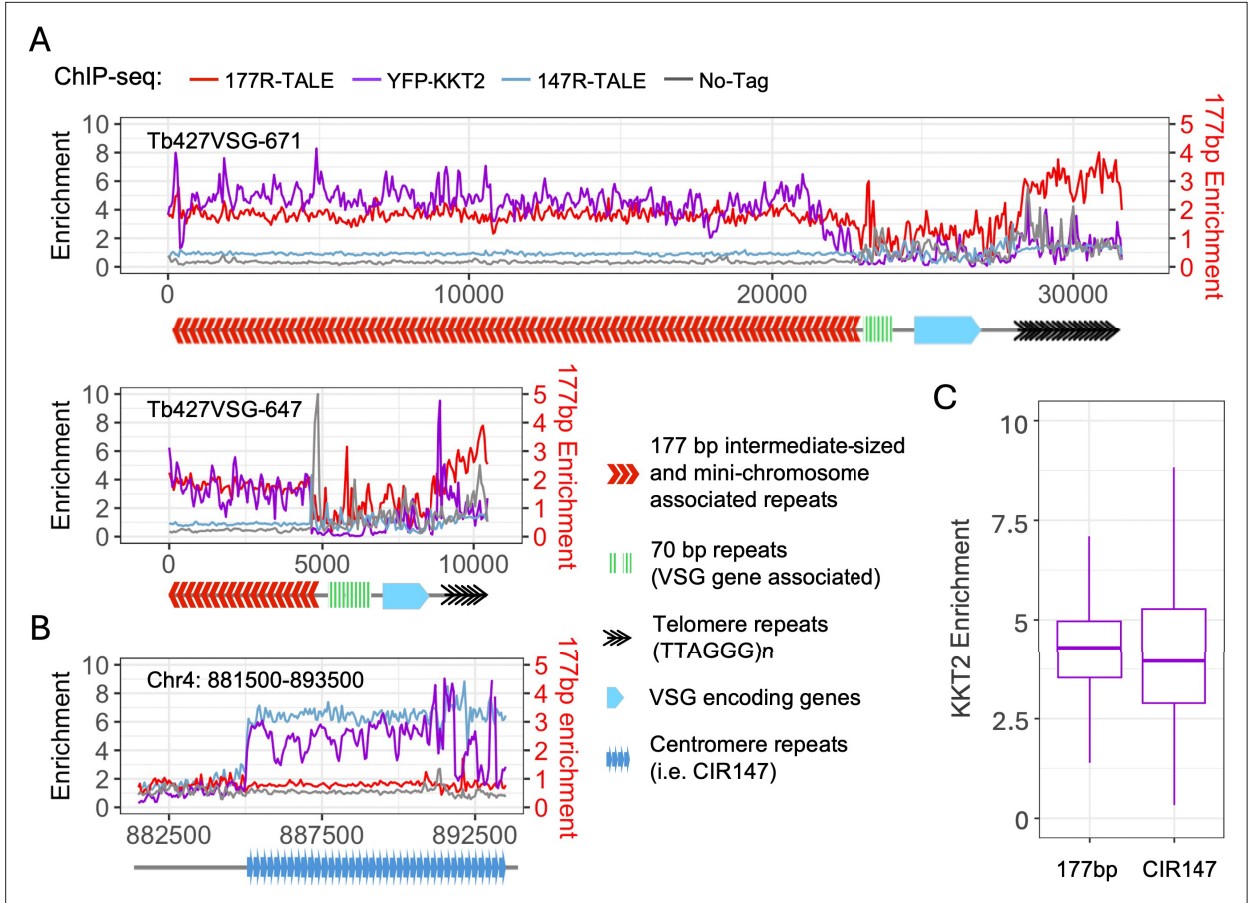

**Figure 7.** Synthetic 177R-TALE-YFP and YFP-KKT2 kinetochore proteins co-localise over 177 bp repeats located on intermediate-sized and mini-chromosomes but not over centromeric CIR147 repeats where 147R-TALE-YFP binds. (**A**) Distribution of 177R-TALE, YFP-KKT2, and 147R-TALE over two intermediate/mini-chromosome telomeres determined by anti-GFP ChIP-seq. Anti-GFP ChIP-seq of *T. brucei* 427 cells expressing no tagged protein is included as a control. The diagram below ChIP-seq profiles indicates the positions of 177 bp repeats (red chevrons), 70 bp repeats (green bars), VSG encoding genes (blue), and telomere (TTAGGG)$_n$ repeats (black chevrons) within Tb427VSG-671_unitig_Tb427v12:17,836–31,606 (31 kb) and Tb427VSG-647_untig_Tb427v12 (10 kb). (**B**) Comparison of distribution of 177R-TALE, 147R-TALE, and YFP-KKT2 over the chromosome 4 CIR147 centromere repeat array and adjacent unique sequences. Chr4:880,000–895,000 (15 kb) and Tb427VSG-671_unitig_Tb427v12:12,000–27,000 (31 kb). The diagram below ChIP-seq indicates the position of CIR147 repeats. (**C**) Comparison of YFP-KKT2 kinetochore protein enrichment on 177 bp and 147 bp repeats. Data are from two biological replicates. Y axis: log$_2$ values, X axis: repeat types.

with 177R-TALE, along with Aurora B kinase itself (***Supplementary file 1l***; ***D'Archivio and Wickstead, 2017***; ***Nerusheva et al., 2019***; ***Zhou et al., 2019***). Cohesin complex subunits were also present in both 177R-TALE-YFP and YFP-KKT2 affinity purifications, underscoring their expected role in mediating sister-kinetochore cohesion during mitosis. Thus, many KKT kinetochore proteins were found to be enriched with affinity-purified 177R-TALE, which ChIP-seq showed associates with 177 bp repeats but not with KKT2-bound centromere regions on the 11 megabase-sized main chromosomes (i.e. 177R-TALE is not enriched on CIR147 or other centromeric repeats; ***Figure 2B***). This finding suggests that kinetochores with a related composition assemble on all *T. brucei* chromosomes regardless of their size classification and that 177 bp repeats attract kinetochore proteins to the intermediate-sized and mini-chromosomes. To explore this possibility further, our YFP-KKT2 and 177R-TALE ChIP-seq data was compared over two regions from intermediate-sized chromosomes spanning 177 bp repeats to the telomere (***Figure 7A***). The resulting analysis revealed that both YFP-KKT2 and 177R-TALE proteins are enriched on 177 bp repeat arrays but not adjacent non-repetitive sequences. In contrast, YFP-KKT2 and 147R-TALE, but not 177R-TALE, were enriched over main chromosome centromeric 147 bp repeat arrays (***Figure 7B***). Taking into account the relative number of CIR147 and 177 bp repeats in the current *T. brucei* genome (***Cosentino et al., 2021***; ***Rabuffo et al., 2024***), comparative analyses demonstrated that YFP-KKT2 is enriched on both CIR147 and 177 bp repeats (***Figure 7C***).

We conclude that kinetochore proteins assemble on at least a proportion of the individual units within the 177 bp repeat arrays on intermediate-sized and/or mini-chromosomes.

## Discussion

Repetitive elements are a feature of most eukaryotic genomes with major roles in defining centromeres and telomeres, and influencing the expression of nearby genes through the formation of specific types of chromatin (*Allshire and Karpen, 2008*; *Allshire and Madhani, 2018*). Kinetoplastids represent a very distinct early-branching eukaryotic lineage, which has highly divergent histones (*Alsford and Horn, 2004*; *Deák et al., 2023*; *Saha et al., 2021*). Consequently, little is known about the repertoire of proteins that associate with chromatin formed on repetitive elements in these organisms despite their importance in chromosome segregation, telomere maintenance, and immune evasion. Here, we have developed an approach which utilises a collection of synthetic DNA binding proteins designed to bind telomeric $(TTAGGG)_n$ (TelR-TALE), centromeric CIR147 (147R-TALE), ingi-related (ingiR-TALE) dispersed, VSG gene-associated 70 bp (70R-TALE), and mini/intermediate chromosome-specific (177R-TALE) repeats when expressed in bloodstream-form *T. brucei* cells. ChIP-seq demonstrated that all five TALE-based synthetic proteins target the repetitive elements that they were designed to bind. Affinity selection of the TelR-TALE, 70R-TALE, and 177R-TALE proteins identified specific sets of enriched proteins. However, the 147R-TALE and ingiR-TALE failed to identify any enrichment of specific proteins following their affinity selection. Encouragingly, the proteins identified as enriched with TelR-TALE showed significant overlap with those identified following affinity selection of the $(TTAGGG)_n$ telomere binding protein TRF (*Figure 6—figure supplement 1*). All subunits of the RPA complex were highly enriched with the 70R-TALE and many kinetochore proteins were identified in 177R-TALE affinity purifications.

It was initially surprising that no specific proteins were detected as being enriched following either 147R-TALE or ingiR-TALE pulldown. Given that both of these synthetic proteins target their designated target sequence in vivo (*Figure 2B*), it seems likely that this failure is related to the fact that there are fewer target sites for these synthetic proteins to bind to than the TelR-TALE, 70R-TALE, and 177R-TALE proteins, which clearly identified proteins bound in their immediate vicinity in *T. brucei* cells. Thus, although proteins that bind CIR147 or ingi-related repeats in vivo may be present, their level of enrichment may not be sufficient to allow detection above background by proteomic analyses following 147R-TALE or ingiR-TALE pulldown. It is also possible that the binding of the 147R-TALE and ingiR-TALE proteins dislodges a significant proportion of the proteins that normally bind these repeats, thus reducing their enrichment. Regardless, the 147R-TALE and ingiR-TALE proteins were well expressed in *T. brucei* cells, but their affinity selection did not significantly enrich for any relevant proteins. Thus, 147R-TALE and ingiR-TALE provide reassurance for the overall specificity for proteins enriched in TelR-TALE, 70R-TALE, and 177R-TALE affinity purifications.

The TelR-TALE binds telomeric $(TTAGGG)_n$ repeats in vivo and copurifies with a collection of proteins known to function at trypanosome telomeres, therefore demonstrating that a synthetic protein designed to bind a repetitive element can be used to identify other proteins enriched over those repeats (*Figure 6A and B*, *Figure 6—figure supplement 1*). Apart from known telomere binding proteins, the two zinc finger proteins ZC3H39 (Tb927.10.14930) and ZC3H40 (Tb927.10.14950) were enriched in both TelR-TALE-YFP and YFP-TRF affinity selections. The fact that both ZC3H39 and ZC3H40 were enriched by affinity selection with these independent baits – one endogenous (YFP-TRF) and the other a synthetic telomere repeat binding protein (TelR-TALE-YFG) – suggests that at least a proportion of these proteins are present near, and may have some function at, telomeres. Although both ZC3H39 and ZC3H40 have also been shown to be involved in the post-transcriptional regulation of transcripts encoding respiratory chain proteins and located primarily in the cytoplasm, both were identified in a genome-wide RNAi screen for telomeric gene derepression that also selected the VSG expression site regulator VEX1 (*Trenaman et al., 2019*). Hence, in addition to their role in respiratory complex gene regulation, ZC3H39 and ZC3H40 might play some additional role in the regulation of gene expression near telomeres. It is possible that they act through telomerase recruitment at telomeres, the regulation of telomere repeat-containing RNA (TERRA) transcripts that are produced at VSG active telomeres (*Saha et al., 2021*), or engagement of some other regulatory complex associated with telomeres.

Analysis of 70 bp repeat-associated proteins via specific 70R-TALE affinity selection identified RPA1, 2, and 3. This heterotrimeric complex is enriched at single-stranded DNA and associated with DNA damage and double-stranded DNA breaks. The accumulation of these proteins at 70 bp repeats is consistent with the function of these sequences in the initiation of recombination events involved in surface antigen switching, with trypanosomes being unusual in not activating a DNA damage cell cycle checkpoint thereby allowing continued proliferation whilst promoting antigenic diversity (*Glover et al., 2019*). Furthermore, the enrichment of FACT complex components with repeat bound 70R-TALE again highlights 70 bp repeats as an expected focus of recombination events and expression site activity. FACT depletion is known to alleviate repression at these silent VSG expression sites by generating a more open chromatin conformation and reciprocally decreases expression from the active VSG expression site (*Denninger and Rudenko, 2014*).

Kinetoplastid kinetochores are unusual in that they are composed of at least 26 proteins that bear little resemblance to the 40–100 constitutive and transient kinetochore-associated proteins that assemble at conventional eukaryotic centromeres (*Akiyoshi and Gull, 2014*; *Ballmer et al., 2024*; *D'Archivio and Wickstead, 2017*; *Yatskevich et al., 2023*). In *T. brucei*, kinetochores assemble at a single location on both copies of the 11 main diploid chromosomes. The centromeres of chromosomes 4, 5, and 8 contain CIR147 repeat arrays over which kinetochore proteins are enriched while the centromeres of other megabase chromosomes form on less well-characterised repetitive elements (*Akiyoshi and Gull, 2014*; *Echeverry et al., 2012*; *Obado et al., 2007*). In addition, the characterisation of the many *T. brucei* mini- and intermediate chromosomes remains incomplete due to the presence of long tandem 177 bp repeat arrays. Our ChIP-seq analyses of synthetic 177R-TALE-YFP location showed that it associated with 177 bp repeats in vivo but not the adjacent VSG gene regions on mini-chromosomes or any region of the 11 megabase chromosomes (*Figure 7*). The detection of a plethora of kinetochore proteins on 177R-TALE-YFP-bound chromatin indicates that kinetochores or a sub-kinetochore complex also assembles on the 177 bp repeats of mini- and intermediate *T. brucei* chromosomes. Consistent with this finding, some enrichment of YFP-tagged KKT2 and KKT3 was previously detected using a model mini-chromosome assemblage, and depletion of kinetochore proteins was shown to cause aberrant segregation of mini/intermediate chromosomes (*Akiyoshi and Gull, 2014*). If, as previously suggested (*Akiyoshi and Gull, 2014*), mini- and intermediate 177 bp repeat bearing chromosomes segregate by somehow 'hitching a ride' via kietochores that are actually assembled on the main chromosomes, then it might be expected that 177R-TALE-YFP ChIP-seq would register some signal over centromere regions of the main chromosomes; however, no such signal was observed (*Figures 2B, 4 and 7*). The 17 kinetochore proteins (KKT1, 2, 3, 4, 6, 7, 9, 10, 11, 12, 16, 17, 18, 24, KKIP3, KKIP4) detected in 177R-TALE-YFP affinity purifications represent most of the components considered to comprise the core kinetochore but represent only a subset of the 26 known *T. brucei* main structural kinetochore proteins. It is possible that a more rudimentary kinetochore is assembled on mini- and intermediate chromosome 177 repeat arrays and that these are sufficient to mediate their accurate segregation.

Interestingly, although targeting TALE proteins to different repetitive sequences selected components specific to each repeat type, some overlap in the proteins detected was observed. For example, enrichment of telomere-associated proteins was detected in some affinity-selected samples using 177R-TALE-YFP, presumably resulting from the juxtaposition of telomeric repeats and 177 bp repeats on mini-chromosomes. Supporting this, KKT3 was reciprocally detected in samples affinity-selected using YFP-TRF. Similarly, enrichment of both FACT subunits with 177R-TALE may simply reflect the proximity of silent telomeric chromatin on mini-chromosomes, or it may also indicate that FACT contributes to a particular chromatin environment at 177 bp repeats.

Although proteins associated with TALE-YFP fusions targeting telomeric, 70 bp and 177 bp repeats were successfully identified, our analyses suggest that target sequences need to be present in many copies (>1000) in a *T. brucei* genome of ~35 Mb to successfully identify associated proteins. Thus, the 147R-TALE-YFP which targets 440 copies of the canonical centromeric CIR147 repeat resulted in no enrichment of associated proteins, although other parameters may also influence the ability of a sequence-bound TALE-YFP protein to enrich for nearby chromatin-associated proteins. Such parameters may include the affinity that the target chromatin-bound proteins have for the repeat sequence of interest, the relatively low affinity that these chromatin proteins may have for any other chromosomal region and their overall relative abundance (for detailed discussion, see *Gauchier et al., 2020*).

Methods such as CUT&RUN (*Skene and Henikoff, 2017*), which should selectively release only TALE-bound chromatin, followed by affinity selection (similar to CUT&RUN.ChIP *Brahma and Henikoff, 2019*), might improve protein enrichment relative to background and allow identification of proteins associated with less abundant sequences. An alternative to synthetic TALE proteins is to utilise tagged catalytically dead Cas9 targeted to specific sequences via a CRISPR-embedded guide RNA. Fusion of TALE or dCAS9 probes to APEX or BirA* enzymes could also be incorporated to perhaps improve the identification of proteins that reside close to the synthetically targeted DNA binding protein (*Gao et al., 2018*; *Myers et al., 2018*). Cas9/CRISPR systems allow precise genome editing in *T. brucei* (*Rico et al., 2018*; *Vasquez et al., 2018*) such that the development of dCas9-based CRISPR tools may improve the performance of future sequence-targeted proteomics. However, an advantage of TALE protein use is that only a single entity needs to be expressed that directly targets the sequence of interest.

In conclusion, we have successfully deployed TALE-based affinity selection of proteins associated with repetitive sequences in the trypanosome genome. This has provided new information concerning telomere biology, chromosomal segregation mechanisms, and immune evasion strategies employed by these evolutionarily divergent pathogens. As well as providing an orthogonal corroboration of existing knowledge of protein interactions with discrete genomic features, this has provided new entry points to dissect these parasites' chromatin architecture. We anticipate that extension to other kinetoplastid parasites could assist exploration of *Leishmania* genome instability as a response to environmental adaptation where, for example, the highly abundant SIDER family (70-fold more numerous than in *T. brucei*; *Bringaud et al., 2007*) might overcome the copy number limitations of analysing retransposon sequences analysed in our study. Likewise, the 195 nt satellite DNA in *T. cruzi* represents 5–10% of the parasite genome and is sufficiently abundant to allow analysis of associated proteins (*Elias et al., 2003*).

# Materials and methods
## TALEs target sequence design
All synthetic TALE proteins were designed to bind 15 bp target sequences following a T/thymine base as required for the TALEN kit (*Ding et al., 2013*). The design of the mini-chromosomal 177 bp repeat binding TALE was informed by available sequences (*Wickstead et al., 2004*). The ingi repeat TALE was designed to bind a target within the conserved 5' region 79 bp of related transposable elements (*Bringaud et al., 2008*). For the design of the CIR147 binding, TALE published sequences were used as reference (*Obado et al., 2007*; *Patrick et al., 2009*); however, a CIR147 bp target sequence with only one exact match was picked. TALEs were designed, which were predicted to bind the known 70 bp repeats (*Boothroyd et al., 2009*) and terminal $(TTAGGG)_n$ repeats. A control NonR-TALE predicted to have no recognised target in the *T. brucei* genome was designed as follows: BLAST searches were used to identify exact matches in the TREU927 reference genome. Candidate sequences with one or more matches were discarded. Each TALE was assembled using the Musunuru/Cowan TALEN kit protocol (*Ding et al., 2013*) and subsequently placed in a vector that allowed expression in *T. brucei* bloodstream cells as described in the main text.

## Trypanosome cell culture
*T. brucei brucei* Lister 427 bloodstream-form monomorphic cells were used for all experiments. Cell lines were grown at 37°C and 5% $CO_2$ in HMI-9 medium supplemented with 10% Fetal Calf Serum (Gibco), 1% Penicillin-Streptomycin (Gibco), and selective drug(s) when required (*Hirumi and Hirumi, 1989*). Cell cultures were maintained below $3\times10^6$ cells/ml. Phleomycin 2.5 µg/ml was used to select transformants containing the TALE construct BleoR gene.

## Trypanosome transfections
$5\times10^7$ cells were harvested per transfection by centrifugation at 1000×*g*, 10 min. Cells were washed once with 5 ml TbBSF transfection buffer (*Schumann Burkard et al., 2011*) and pelleted again by centrifugation at 1000×*g*, 10 min before resuspending in 100 µl ice-cold TbBSF transfection buffer, and transferred to an electroporation cuvette (Ingenio). 10–20 µl of DNA for transfection containing 1–5 µg DNA was added to the cuvette. Cells were electroporated in the Amaxa Nucleofector II

(Lonza) using the X-001 programme for bloodstream cells. A 'no DNA' mock transfection was always performed in parallel as a negative control. Electroporated bloodstream cells were added to 30 ml HMI-9 medium and two 10-fold serial dilutions were performed in order to isolate clonal Phleomycin-resistant populations from the transfection. 1 ml of transfected cells was plated per well on 24-well plates (1 plate per serial dilution) and incubated at 37°C and 5% $CO_2$ for a minimum of 6 hr before adding 1 ml media containing 2× concentration Phleomycin (5 μg/ml) per well. A positive control was also performed by adding media containing no selective drug to 12 wells of the control transfection plate.

## Western analyses

Cells were harvested by centrifugation at 1000×*g*, 10 min, washed with 1× PBS and resuspended in 1× PBS + 4× NuPAGE LDS Sample Buffer (Thermo Fisher Scientific) to give a final concentration of $5 \times 10^6$ cells per 10 μl. Samples were then boiled at 95°C for 5 min to ensure cells were dead before removal from the CAT3 facility. Samples were then subjected to sonication using a Diagenode Bioruptor for 10 cycles, 30 s ON/30 s OFF at 4°C on high setting to shear the DNA and reduce the viscosity to aid loading on gels. Samples were run on NuPAGE Bis-Tris Mini Gels (Thermo Fisher Scientific) in a Mini Gel Tank (Thermo Fisher Scientific) in 1× NuPAGE MES Running Buffer at 200 V. Following PAGE, proteins were transferred onto nitrocellulose membranes in a Mini Blot Module (Thermo Fisher Scientific) at 20 V for 1 hr. Membranes were stained with Ponceau S (Sigma-Aldrich) to assess efficiency of protein transfer. After blocking with 5% milk/PBS-T (PBS + 0.05% Tween), membranes were incubated with mouse anti-GFP (Roche) (1:1000 in 5% milk in PBS-T) or anti-BB2 antibody (Hybrydome mouse monoclonal, clone BB2) (1:5 in 5% milk in PBS-T) at 4°C overnight on a lab rocker, then washed with PBS-T and incubated with HRP-conjugated anti-mouse secondary antibody (1:2500 in 5% milk in PBS-T) at room temperature for 1 hr. Membranes were washed with PBS-T and incubated with Amersham ECL Prime Western Blotting Detection Reagent (GE Healthcare) following the manufacturer's instructions. Proteins were visualised using Amersham Hyperfilm ECL (GE Healthcare).

## Fluorescent immunolocalisation

Cells were fixed with 4% paraformaldehyde for 10 min on ice. Fixation was stopped with 0.1 M glycine. Cells were added to polylysine-coated slides and permeabilised with 0.1% Triton X-100. The slides were blocked with 2% BSA. Rabbit anti-GFP primary antibody (Thermo Fisher Scientific A-11122) was used at 1:500 dilution, and secondary Alexa Fluor-568 or -488 goat antirabbit antibody (Thermo Fisher Scientific) was used at 1:1000 dilution. Images were taken with a Zeiss Axio Imager microscope.

## Chromatin immunoprecipitation and sequencing

As previously described (*Staneva et al., 2021*), $3.5 \times 10^8$ parasites were fixed with 0.8% formaldehyde for 20 min at room temperature. Cells were lysed and sonicated in the presence of 0.2% SDS for 30 cycles (30 s ON, 30 s OFF) using the high setting on a Bioruptor sonicator (Diagenode). Cell debris was pelleted by centrifugation, and SDS in the lysate supernatants was diluted to 0.07%. Input samples were taken before incubating the rest of the cell lysates overnight with 10 μg rabbit anti-GFP antibody (Thermo Fisher Scientific A-11122) and Protein G Dynabeads. The beads were washed, and the DNA eluted from them was treated with RNase and Proteinase K. DNA was then purified using a QIAquick PCR purification kit (QIAGEN), and libraries were prepared using NEXTflex barcoded adapters (Bio Scientific). The libraries were sequenced on Illumina NextSeq (Western General Hospital, Edinburgh). In all cases, 75 bp paired-end sequencing was performed. Our subsequent analyses were based on two replicates for all TALEs.

## ChIP-seq data analysis

Sequencing data were mapped to the Tb427V12 genome build (*Rabuffo et al., 2024*) using Bowtie2 (version 2.4.2), with duplicate reads removed using SAMtools (*Danecek et al., 2021*). The default mode of Bowtie 2 was used, which searches for multiple alignments and reports the best one or, if several alignments are deemed equally good, reports one of those randomly. The peaks were identified using MACS2 (version 2.2.7.1) broad peak call. The ChIP samples were normalised to their respective inputs (ratio of ChIP to input reads) and the genome overview was generated using deepTools (*Ramírez et al., 2016*) with 5 bp sliding window.

## Background enrichment calculation

The genome was divided into 50 bp sliding windows, and each window was annotated based on overlapping genomic features, including CIR147, 177 bp repeats, 70 bp repeats, and telomeric $(TTAGGG)_n$ repeats. Windows that did not overlap with any of these annotated repeat elements were defined as 'background' regions and used to establish the baseline ChIP-seq signal. Enrichment for each window was calculated using bamCompare, as $\log_2(IP/Input)$. To adjust for background signal amongst all samples, enrichment values for each sample were further normalised against the corresponding No-YFP ChIP-seq dataset.

## Affinity purification and LC-MS/MS proteomic analysis

As previously described (*Staneva et al., 2021*), cells, $3.5 \times 10^8$, were lysed per IP in the presence of 0.2% NP-40 and 150 mM KCl. Lysates were sonicated briefly (three cycles, 12 s ON, 12 s OFF) at a high setting in a Bioruptor (Diagenode) sonicator. The soluble and insoluble fractions were separated by centrifugation, and the soluble fraction was incubated for 1 hr at 4°C with beads cross-linked to mouse anti-GFP antibody (Roche 11814460001). The resulting immunoprecipitates were washed three times with lysis buffer, and protein was eluted with RapiGest SF Surfactant (Waters) for 15 min at 55°C. Next, filter-aided sample preparation (FASP) (*Wiśniewski et al., 2009*) was used to digest the protein samples for mass spectrometric analysis. Briefly, proteins were reduced with DTT and then denatured with 8 M urea in Vivacon spin (filter) column 30 K cartridges. Samples were alkylated with 0.05 M IAA and digested with 0.5 µg MS-grade Pierce trypsin protease (Thermo Fisher Scientific) overnight, desalted using stage tips (*Rappsilber et al., 2007*), and resuspended in 0.1% TFA for LC-MS/MS. Peptides were separated using RSLC Ultimate 3000 system (Thermo Fisher Scientific) fitted with an EasySpray column (50 cm; Thermo Fisher Scientific) using 2%–40%–95% nonlinear gradients with solvent A (0.1% formic acid) and solvent B (80% acetonitrile in 0.1% formic acid). The EasySpray column was directly coupled to an Orbitrap Fusion Lumos (Thermo Fisher Scientific) operated in DDA mode. 'TopSpeed' mode was used with 3 s cycles with standard settings to maximise identification rates: MS1 scan range 350–1500 mz, RF lens 30%, AGC target 4.0e5 with intensity threshold 5.0e3, filling time 50 ms and resolution 120,000, monoisotopic precursor selection, and filter for charge states 2–5.

HCD (27%) was selected as fragmentation mode. MS2 scans were performed using an ion trap mass analyser operated in rapid mode with AGC set to 2.0e4 and filling time to 50 ms. The dynamic exclusion was set at 60 s.

The MaxQuant software platform (*Cox and Mann, 2008*) version 1.6.1.0 was used to process the raw files, and search was conducted against *T. brucei brucei* complete/reference proteome (Uniprot – released in April 2019), using the Andromeda search engine (*Cox et al., 2011*). For the first search, peptide tolerance was set to 20 ppm, while for the main search, it was set to 4.5 ppm. The isotope mass tolerance was set to 2 ppm, with a maximum charge of 7. Digestion mode was set to 'specific' with trypsin, allowing a maximum of two missed cleavages. Carbamidomethylation of cysteine was set as a fixed modification. Oxidation of methionine was set as a variable modification. Label-free quantitation analysis was performed by employing the MaxLFQ algorithm as described by *Cox et al., 2014*. Absolute protein quantification was performed as described in *Schwanhäusser et al., 2011*. Peptide and protein identifications were filtered to 1% FDR. Statistical analysis and visualisation were performed using Perseus version 1.6.2.1 (*Tyanova et al., 2016*).

## Acknowledgements

The authors thank Alison Pidoux for comments on the manuscript and assistance with images, Bungo Akiyoshi for comments and discussion, and Shaun Webb of the Wellcome Centre for Cell Biology and Discovery Research Platform for Hidden Cell Biology Bioinformatics Core for maintaining servers and pipelines for processing sequencing data. The authors also thank Julie Young for laboratory management support for trypanosome culture during this project. This work was funded by a UKRI/BBSRC EastBio PhD studentship supporting Tadhg Devlin (BB/M010996/1), an MRC Research Grant awarded to RCA and KRM and supporting RC (MR/T04702X/1), a Wellcome Investigator Award to KRM (221717), a Wellcome Principal Research Fellowship to RCA supporting RC and TA (200885; 224358), a Wellcome Instrument grant to JR (108504), and core funding for the Wellcome Centre for Cell Biology (203149) and subsequently the Wellcome funded Discovery Research Platform for

Hidden Cell Biology DRP-HCB supporting CS (226791). For the purpose of Open Access, the authors have applied a CC BY public copyright licence to any Author Accepted Manuscript version arising from this submission.

## Additional information

### Funding

| Funder | Grant reference number | Author |
| --- | --- | --- |
| Biotechnology and Biological Sciences Research Council | BB/M010996/1 | Tadhg Devlin |
| Medical Research Council | MR/T04702X/1 | Keith R Matthews Robin C Allshire |
| Wellcome | 10.35802/221717 | Keith R Matthews |
| Wellcome | 10.35802/200885 | Robin C Allshire |
| Wellcome | 10.35802/224358 | Robin C Allshire |
| Wellcome | 10.35802/108504 | Juri Rappsilber |
| Wellcome | 10.35802/203149 | Robin C Allshire |
| Wellcome | 10.35802/226791 | Robin C Allshire |

The funders had no role in study design, data collection and interpretation, or the decision to submit the work for publication. For the purpose of Open Access, the authors have applied a CC BY public copyright license to any Author Accepted Manuscript version arising from this submission.

### Author contributions

Roberta Carloni, Conceptualization, Formal analysis, Investigation, Visualization, Methodology, Writing – original draft, Writing – review and editing; Tadhg Devlin, Conceptualization, Formal analysis, Investigation, Methodology, Writing – original draft; Pin Tong, Data curation, Formal analysis, Investigation, Visualization, Methodology, Writing – original draft, Writing – review and editing; Christos Spanos, Data curation, Formal analysis, Investigation, Visualization, Methodology; Tanya Auchynnikava, Formal analysis, Methodology; Juri Rappsilber, Funding acquisition; Keith R Matthews, Robin C Allshire, Conceptualization, Supervision, Funding acquisition, Visualization, Writing – original draft, Project administration, Writing – review and editing

### Author ORCIDs

Christos Spanos ⓘ https://orcid.org/0000-0002-4376-8242
Juri Rappsilber ⓘ https://orcid.org/0000-0001-5999-1310
Keith R Matthews ⓘ https://orcid.org/0000-0003-0309-9184
Robin C Allshire ⓘ https://orcid.org/0000-0002-8005-3625

Reviewer #1 (Public review): https://doi.org/10.7554/eLife.109950.2.sa1
Reviewer #2 (Public review): https://doi.org/10.7554/eLife.109950.2.sa2
Author response https://doi.org/10.7554/eLife.109950.2.sa3

## Additional files

### Supplementary files

Supplementary file 1. Proteomics analyses comparing protein enrichments in the indicated affinity selections a-to-q. (a) Affinity selection data for wild-type cells expressing No YFP versus cells expressing YFP-TRF (WT NoYFP vs. YFP-TRF). Proteins enriched in YFP-TRF affinity selections were identified and quantified by LC-MS/MS analysis relative to affinity selections from wild-type cells lacking any YFP as a negative control. (b) Affinity selection data for wild-type cells expressing

No YFP versus cells expressing TelR-TALE (WT NoYFP vs. TelR-TALE). Proteins enriched in TelR-TALE affinity selections were identified and quantified by LC-MS/MS analysis relative to affinity selections from wild-type cells lacking any YFP as a negative control. (c) Affinity selection data for wild-type cells expressing No YFP versus cells expressing NonR-TALE (WT NoYFP vs. NonR-TALE). Proteins enriched in NonR-TALE affinity selections were identified and quantified by LC-MS/MS analysis relative to affinity selections from wild-type cells lacking any YFP as a negative control. (d) Affinity selection data for cells expressing NonR-TALE versus cells expressing TelR-TALE (NonR-TALE vs. TelR-TALE). Proteins enriched in TelR-TALE affinity selections were identified and quantified by LC-MS/MS analysis relative to affinity selections of NonR-TALE. (e) Affinity selection data for wild-type cells expressing No YFP versus cells expressing 147R-TALE (WT NoYFP vs. 147R-TALE). Proteins enriched in 147R-TALE affinity selections were identified and quantified by LC-MS/MS analysis relative to affinity selections from wild-type cells lacking any YFP as a negative control. (f) Affinity selection data for cells expressing NonR-TALE versus cells expressing 147R-TALE (NonR-TALE vs. TelR-TALE). Proteins enriched in 147R-TALE affinity selections were identified and quantified by LC-MS/MS analysis relative to affinity selections of NonR-TALE. (g) Affinity selection data for wild-type cells expressing No YFP versus cells expressing ingiR-TALE (WT NoYFP vs. ingiR-TALE). Proteins enriched in ingiR-TALE affinity selections were identified and quantified by LC-MS/MS analysis relative to affinity selections from wild-type cells lacking any YFP as a negative control. (h) Affinity selection data for cells expressing NonR-TALE versus cells expressing ingiR-TALE (NonR-TALE vs. ingiR-TALE). Proteins enriched in ingiR-TALE affinity selections were identified and quantified by LC-MS/MS analysis relative to affinity selections of NonR-TALE. (i) Affinity selection data for wild-type cells expressing No YFP versus cells expressing 70R-TALE (WT NoYFP vs. 70R-TALE). Proteins enriched in 70R-TALE affinity selections were identified and quantified by LC-MS/MS analysis relative to affinity selections from wild-type cells lacking any YFP as a negative control. (j) Affinity selection data for wild-type cells expressing No YFP versus cells expressing YFP-RPA2 (WT NoYFP vs. YFP-RPA2). Proteins enriched in YFP-RPA2 affinity selections were identified and quantified by LC-MS/MS analysis relative to affinity selections from wild-type cells lacking any YFP as a negative control. (k) Affinity selection data for cells expressing NonR-TALE versus cells expressing 70R-TALE (NonR-TALE vs. 70R-TALE). Proteins enriched in 70R-TALE affinity selections were identified and quantified by LC-MS/MS analysis relative to affinity selections of NonR-TALE. (l) Affinity selection data for wild-type cells expressing No YFP versus cells expressing 177R-TALE (WT NoYFP vs. 177R-TALE). Proteins enriched in 177R-TALE affinity selections were identified and quantified by LC-MS/MS analysis relative to affinity selections from wild-type cells lacking any YFP as a negative control. (m) Affinity selection data for cells expressing NonR-TALE versus cells expressing 177R-TALE (NonR-TALE vs. 177R-TALE). Proteins enriched in 177R-TALE affinity selections were identified and quantified by LC-MS/MS analysis relative to affinity selections of NonR-TALE. (n) Affinity selection data for wild-type cells expressing No YFP versus cells expressing YFP-RPA2 (WT NoYFP vs. YFP-KKT2). Proteins enriched in YFP-KKT2 affinity selections were identified and quantified by LC-MS/MS analysis relative to affinity selections from wild-type cells lacking any YFP as a negative control. (o) Affinity selection data for cells expressing ingiR-TALE versus cells expressing TelR-TALE (ingiR-TALE vs. TelR-TALE). Proteins enriched in TelR-TALE affinity selections were identified and quantified by LC-MS/MS analysis relative to affinity selections of ingiR-TALE. (p) Affinity selection data for cells expressing ingiR-TALE versus cells expressing 70R-TALE (ingiR-TALE vs. 70R-TALE) Proteins enriched in 70R-TALE affinity selections were identified and quantified by LC-MS/MS analysis relative to affinity selections of ingiR-TALE. SuppFile1p_ingiR-TALEv70R-TALE (q) Affinity selection data for cells expressing ingiR-TALE versus cells expressing 177R-TALE (ingiR-TALE vs. 70R-TALE) Proteins enriched in 177R-TALE affinity selections were identified and quantified by LC-MS/MS analysis relative to affinity selections of ingiR-TALE. SuppFile1q_ingiR-TALEv177R-TALE.

MDAR checklist

## Data availability

Sequence Data: All NGS ChIP-seq data generated have been submitted to and will be available under an accession number at the NCBI Gene Expression Omnibus (GEO; https://www.ncbi.nlm.nih.gov/geo/). The GEO accession number for ChIP-seq data is: GSE295698. Proteomics Data: All LC-MS/MS proteomics data generated are available on the Proteomics Identification Database (PRIDE; https://www.ebi.ac.uk/pride/archive/projects/PXD063130) with accession number PXD063130.

The following datasets were generated:

| Author(s) | Year | Dataset title | Dataset URL | Database and Identifier |
|---|---|---|---|---|
| Carloni R, Devlin T, Tong P, Auchynnikava T, Spanos CR, Rappsilber J, Matthews KR, Allshire RC | 2025 | Defining the chromatin-associated protein landscapes on *Trypanosoma brucei* repetitive elements using synthetic TALE proteins | https://www.ncbi.nlm.nih.gov/geo/query/acc.cgi?acc=GSE295698 | NCBI Gene Expression Omnibus, GSE295698 |
| Carloni R, Devlin T, Tong P, Auchynnikava T, Spanos CR, Rappsilber J, Matthews KR, Allshire RC | 2025 | Defining the chromatin-associated protein landscapes on *Trypanosoma brucei* repetitive elements using synthetic TALE proteins | https://www.ebi.ac.uk/pride/archive/projects/PXD063130 | PRIDE, PXD063130 |

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

# Appendix 1

## Appendix 1—key resources table

| Reagent type (species) or resource | Designation | Source or reference | Identifiers | Additional information |
|---|---|---|---|---|
| Gene (*T. brucei*) | TRF gene | TriTrypDB | Tb927.11.1000 | Telomere Repeat binding Factor |
| Gene (*T. brucei*) | KKT2 gene | TriTrypDB | Tb927.10.12850 | Kinetoplastid KineTochore protein 2 |
| Strain background (*T. brucei* Lister 427) | *T. brucei* Lister 427 | Standard | RRID:CVCL_K226 | Bloodstream form, monomorphic |
| Cell line (*T. brucei*) | *T. brucei* Lister 427 bloodstream form | Standard laboratory strain | RRID:CVCL_K226 | Monomorphic strain, used as parental line |
| Cell line (*T. brucei*) | YFP-TRF | *Akiyoshi and Gull, 2014*. *Staneva et al., 2021* | N/A | *T. brucei* 427 expressing YFP-tagged TRF (Tb927.10.12850) |
| Cell line (*T. brucei*) | YFP-KKT2 | *Akiyoshi and Gull, 2014*. *Staneva et al., 2021* | N/A | *T. brucei* 427 expressing YFP-tagged KKT2 (Tb927.10.12850) |
| Cell line (*T. brucei*) | TelR-TALE-YFP | This paper | N/A | *T. brucei* 427 expressing TelR-TALE-YFP |
| Cell line (*T. brucei*) | 70R-TALE-YFP | This paper | N/A | *T. brucei* 427 expressing 70R-TALE-YFP |
| Cell line (*T. brucei*) | 147R-TALE-YFP | This paper | N/A | *T. brucei* 427 expressing 147R-TALE-YFP |
| Cell line (*T. brucei*) | 177R-TALE-YFP | This paper | N/A | *T. brucei* 427 expressing 177R-TALE-YFP |
| Cell line (*T. brucei*) | ingiR-TALE-YFP | This paper | N/A | *T. brucei* 427 expressing ingiR-TALE-YFP |
| Cell line (*T. brucei*) | NonR-TALE-YFP | This paper | N/A | *T. brucei* 427 expressing NonR-TALE-YFP |
| Transfected construct (*E. coli*) | pTALE-TelR | This paper | De novo construct | Plasmid to generate TelR-TALE-YFP cell line |
| Transfected construct (*E. coli*) | pTALE-70R | This paper | De novo construct | Plasmid to generate 70R-TALE-YFP cell line |
| Transfected construct (*E. coli*) | pTALE-147R | This paper | De novo construct | Plasmid to generate 147R-TALE-YFP cell line |
| Transfected construct (*E. coli*) | pTALE-177R | This paper | De novo construct | Plasmid to generate 177R-TALE-YFP cell line |
| Transfected construct (*E. coli*) | pTALE-ingiR | This paper | De novo construct | Plasmid to generate ingiR-TALE-YFP cell line |
| Transfected construct (*E. coli*) | pTALE-NonR | This paper | De novo construct | Plasmid to generate NonR-TALE-YFP cell line |
| Recombinant DNA reagent | pPOTv4 TRF fusion PCR | As in *Staneva et al., 2021* | N/A | For endogenous YFP-tagging of TRF |
| Recombinant DNA reagent | pPOTv4 KKT2 fusion PCR | As in *Staneva et al., 2021* | N/A | For endogenous YFP-tagging of KKT2 |
| Recombinant DNA reagent | pPOTv4 RPA2 fusion PCR | As in *Staneva et al., 2021* | N/A | For endogenous YFP-tagging of RPA2 |
| Antibody | Mouse anti-Ty1 (BB2) (Monoclonal) | Thermo Fisher Scientific | Cat# MA5-23513; RRID:AB_2610643 | (1:5) for western blots |
| Antibody | Mouse anti-GFP (Monoclonal) | Roche | Cat# 11814460001; RRID:AB_390913 | Used for Affinity Purification (beads cross-linked) |
| Antibody | Mouse anti-GFP (Monoclonal) | Roche | Cat# 11814460001;RRID:AB_390913 | (1:1000) for western blots |
| Antibody | Goat anti-mouse IgG (H+L) Alexa Fluor 568 | Thermo Fisher Scientific | Cat# A-11004; RRID:AB_2534072 | (1:1000) for IF |
| Antibody | Goat anti-mouse IgG (H+L) HRP-conjugated | Thermo Fisher Scientific | Cat# 31430; RRID:AB_228307 | (1:5000) for western blot |
| Sequence-based reagent | TelR-TALE target sequence | This paper | N/A | Designed to bind AGGGTTAGGGTTAGG. TALE in vivo truncation recognises AGGGTTAG |
| Sequence-based reagent | 70R-TALE target sequence | This paper | N/A | AGGAGAGTGTTGTGA |
| Sequence-based reagent | 147R-TALE target sequence | This paper | N/A | GCAGCGTTGTGCATG |
| Sequence-based reagent | ingiR-TALE target sequence | This paper | N/A | GCCGGCCACCTCAAC |
| Sequence-based reagent | NonR-TALE target sequence | This paper | N/A | GGAAGTATACCTGGC (no genomic match) |

*Appendix 1 Continued on next page*

*Appendix 1 Continued*

| Reagent type (species) or resource | Designation | Source or reference | Identifiers | Additional information |
|---|---|---|---|---|
| Sequence-based reagent | TALE-PCR-LP (Primer) | This paper | N/A | Forward primer for integration check (*Figure 1—figure supplement 1*) |
| Sequence-based reagent | TALE-PCR-RP (Primer) | This paper | N/A | Reverse primer for integration check (see *Figure 1—figure supplement 1*) |
| Peptide, recombinant protein | Protein G Dynabeads | Thermo Fisher Scientific | Cat# 10004D | Used for ChIP |
| Peptide, recombinant protein | Trypsin Protease, MS Grade | Thermo Fisher Scientific (Pierce) | Cat# 90057 | Used for protein digestion |
| Commercial assay or kit | TALEN module kit | *Ding et al., 2013*. | N/A | Used for TALE assembly |
| Commercial assay or kit | NEXTflex barcoded adapters | Bio Scientific | N/A | Used for library preparation |
| Commercial assay or kit | NuPAGE Bis-Tris Mini Gels | Thermo Fisher Scientific | Cat# NP0321BOX | For protein separation |
| Commercial assay or kit | NuPAGE LDS Sample Buffer (4×) | Thermo Fisher Scientific | Cat# NP0007 | For western blot sample preparation |
| Commercial assay or kit | Amersham ECL Prime | GE Healthcare | Cat# RPN2232 | Western blot detection |
| Commercial assay or kit | Amersham Hyperfilm ECL | GE Healthcare | Cat# 28906839 | Film for western blot visualisation |
| Commercial assay or kit | Vivacon 500 (30 K MWCO) | Sartorius | Cat# VN01H22 | Spin filters used for FASP protocol |
| Chemical compound, drug | RapiGest SF Surfactant | Waters | Cat# 186001861 | Used for protein elution in AP-MS |
| Chemical compound, drug | HMI-9 medium | Standard | N/A | For *T. brucei* culture |
| Chemical compound, drug | Fetal Calf Serum (FCS) | Gibco (Thermo Fisher Scientific) | Cat# 10500064 | 10% supplement for HMI-9 |
| Chemical compound, drug | Ponceau S | Sigma-Aldrich | Cat# P3504 | Membrane staining |
| Chemical compound, drug | Dithiothreitol (DTT) | Sigma-Aldrich (or similar) | Cat# D0632 | Reducing agent |
| Chemical compound, drug | Iodoacetamide (IAA) | Sigma-Aldrich (or similar) | Cat# I1149 | Alkylating agent |
| Chemical compound, drug | Urea | Sigma-Aldrich (or similar) | Cat# U5378 | Denaturing agent (8 M) |
| Chemical compound, drug | Bovine Serum Albumin (BSA) | Sigma-Aldrich (or similar) | N/A | Blocking agent (2%) for IF |
| Chemical compound, drug | Triton X-100 | Sigma-Aldrich (or similar) | N/A | Permeabilisation (0.1%) for IF |
| Chemical compound, drug | Paraformaldehyde | Sigma-Aldrich (or similar) | N/A | Fixation (4%) for IF |
| Chemical compound, drug | Blasticidin S | *InvivoGen* or similar | Cat# ant-bl-1 | (10 µg/ml) Used for TALE-YFP selection |
| Software, algorithm | MaxQuant | *Cox and Mann, 2008* | RRID:SCR_014485 | v.2.0.3.0 used for proteomic analysis |
| Software, algorithm | Perseus | *Tyanova et al., 2016* | RRID:SCR_015753 | v.1.6.15.0 used for proteomic statistical analysis |
| Software, algorithm | Bowtie2 | *Langmead and Salzberg, 2012* | RRID:SCR_016368 | v.2.4.2 used for ChIP-seq alignment |
| Software, algorithm | MACS2 | *Zhang et al., 2008* | | v.2.2.7.1 used for peak calling |
| Software, Algorithm | SAMtools | *Danecek et al., 2021* | RRID:SCR_002105 | Used for removing duplicate reads |
| Software, algorithm | deepTools | *Ramírez et al., 2016* | RRID:SCR_016366 | Used for genome overview |
| Other | Orbitrap Fusion Lumos | Thermo Fisher Scientific | N/A | Mass spectrometer used |
| Other | Zeiss Axio Imager | Zeiss | N/A | Microscope used for imaging |
| Other | Illumina NextSeq 500/550 | Illumina | N/A | Used for ChIP-seq library sequencing |
| Other (software, algorithm) | TriTrypDB | https://tritrypdb.org | RRID:SCR_007043 | Database used for *T. brucei* gene identifiers |

