## [Editor Report · eLife Assessment]

This work significantly advances our understanding of chromatin organization within regions of repetitive sequences in the parasitic protozoan *Trypanosoma brucei*. Using cutting edge interdisciplinary tools, the authors provide **compelling** evidence for two discrete types of repetitive DNA element-associated proteins- one set involved in essential centromere function; and, the other involved in glycoprotein antigenic variation via homologous recombination. Thus, these **fundamental** findings have implications for this parasite's biology, and for therapeutic targeting in kinetoplastid diseases. This work will be exciting to those in the centromere/mitosis and parasite immunity fields.

[Editors' note: this paper was reviewed by Review Commons.]

---

## [Referee Report · Reviewer #1 (Public review)]

Summary:

Carloni et al. comprehensively analyze which proteins bind repetitive genomic elements in *Trypanosoma brucei*. For this, they perform mass spectrometry on custom-designed, tagged programmable DNA-binding proteins. After extensively verifying their programmable DNA-binding proteins (using bioinformatic analysis to infer target sites, microscopy to measure localization, ChIP-seq to identify binding sites), they present, among others, two major findings: (1) 14 of the 25 known *T. brucei* kinetochore proteins are enriched at 177bp repeats. As *T. brucei*'*s* 177bp repeat-containing intermediate-sized and mini-chromosomes lack centromere repeats but are stable over mitosis, Carloni et al. use their data to hypothesize that a 'rudimentary' kinetochore assembles at the 177bp repeats of these chromosomes to segregate them. (2) 70bp repeats are enriched with the Replication Protein A complex, which, notably, is required for homologous recombination. Homologous recombination is the pathway used for recombination-based antigenic variation of the 70bp-repeat-adjacent variant surface glycoproteins.

Strengths and Weaknesses:

The manuscript was previously reviewed through Review Commons. As noted there, the experiments are well controlled, the claims are well supported, and the methods are clearly described. The conclusions are convincing. All concerns I raised have been addressed except one (minor point #8):

"The way the authors mapped the ChIP-seq data is potentially problematic when analyzing the same repeat type in different genomic regions. Reads with multiple equally good mapping positions were assigned randomly. This is fine when analyzing repeats by type, independent of genomic position, which is what the authors do to reach their main conclusions. However, several figures (Fig. 3B, Fig. 4B, Fig. 5B, Fig. 7) show the same repeat type at specific genomic locations." Due to the random assignment, all of these regions merely show the average signal for the given repeat. I find it misleading that this average is plotted out at "specific" genomic regions.

Initially, I suggested a workaround, but the authors clarified why the workaround was not feasible, and their explanation is reasonable to me. That said, the figures still show a signal at positions where they can't be sure it actually exists. If this cannot be corrected analytically, it should at least be noted in the figure legends, Results, or Discussion.

Importantly, the authors' conclusions do not hinge on this point; they are appropriately cautious, and their interpretations remain valid regardless.

Significance:

This work is of high significance for chromosome/centromere biology, parasitology, and the study of antigenic variation. For chromosome/centromere biology, the conceptual advancement of different types of kinetochores for different chromosomes is a novelty, as far as I know. It would certainly be interesting to apply this study as a technical blueprint for other organisms with mini-chromosomes or chromosomes without known centromeric repeats. I can imagine a broad range of labs studying other organisms with comparable chromosomes to take note of and build on this study. For parasitology and the study of antigenic variation, it is crucial to know how intermediate- and mini-chromosomes are stable through cell division, as these chromosomes harbor a large portion of the antigenic repertoire. Moreover, this study also found a novel link between the homologous repair pathway and variant surface glycoproteins, via the 70bp repeats. How and at which stages during the process, 70bp repeats are involved in antigenic variation is an unresolved, and very actively studied, question in the field. Of course, apart from the basic biological research audience, insights into antigenic variation always have the potential for clinical implications, as *T. brucei* causes sleeping sickness in humans and nagana in cattle. Due to antigenic variation, *T. brucei* infections can be chronic.

Comments on revised version:

All my recommendations have been addressed.

---

## [Referee Report · Reviewer #2 (Public review)]

The *Trypanosoma brucei* genome, like that of other eukaryotes, contains diverse repetitive elements. Yet, the chromatin-associated proteome of these regions remains largely unexplored. This study represents a very important conceptual and technical advancement by employing synthetic TALE DNA-binding proteins fused to YFP to selectively capture proteins associated with specific repetitive sequences in *T. brucei* chromatin. The data presented here are convincing, supported by appropriate controls and a well-validated methodology, aligned with current state-of-the-art approaches.

The authors used synthetic TALE DNA binding proteins, tagged with YFP, which were designed to target five specific repeat elements in *T. brucei* genome, including centromere and telomeres-associated repeats and those of a transposon element. This is in order to identify specific proteins that bind to these repetitive sequences in *T. brucei* chromatin. Validation of the approach was done using a TALE protein designed to target the telomere repeat (TelR-TALE) that detected many of the proteins that were previously implicated with telomeric functions. A TALE protein designed to target the 70 bp repeats that reside adjacent to the VSG genes (70R-TALE) detected proteins that function in DNA repair and a protein designed to target the 177 bp repeat arrays (177R-TALE) identified kinetochore proteins associated *T. brucei* mega base chromosomes, as well as in intermediate and mini-chromosomes, which imply that kinetochore assembly and segregation mechanisms are similar in all *T. brucei* chromosomes.

This study represents a significant conceptual and technical advancement. To the best of our knowledge, it is the first report of employing TALE-YFP for affinity-based detection of protein complexes bound to repetitive genomic sequences in *T. brucei*. This approach enhances our understanding the organization in these important regions of the trypanosomal chromatin and provides the foundation for investigating the functional roles of associated proteins in parasite biology. These findings will be of particular interest to researchers studying the molecular biology of kinetoplastid parasites and other unicellular organisms, as well as to scientists investigating the roles of repetitive genomic elements in chromatin structure and their functional role in higher eukaryotes.

Importantly, any essential or unique interacting partners identified using the approach employed here, could serve as a potential target for therapeutic intervention in severe tropical diseases cause by kinetoplastids.

---

## [Author Response]

**Point-by-point description of the revisions:**

**Reviewer #1 (Evidence, reproducibility and clarity):**
SummaryIn this article, the authors used the synthetic TALE DNA binding proteins, tagged with YFP, which were designed to target five specific repeat elements in *Trypanosoma brucei* genome, including centromere and telomeres-associated repeats and those of a transposon element. This is in order to detect and identified, using YFP-pulldown, specific proteins that bind to these repetitive sequences in *T. brucei* chromatin. Validation of the approach was done using a TALE protein designed to target the telomere repeat (TelR-TALE) that detected many of the proteins that were previously implicated with telomeric functions. A TALE protein designed to target the 70 bp repeats that reside adjacent to the VSG genes (70R-TALE) detected proteins that function in DNA repair and the protein designed to target the 177 bp repeat arrays (177R-TALE) identified kinetochore proteins associated *T. brucei* mega base chromosomes, as well as in intermediate and mini-chromosomes, which imply that kinetochore assembly and segregation mechanisms are similar in all *T. brucei* chromosome.Major comments:Are the key conclusions convincing?The authors reported that they have successfully used TALE-based affinity selection of proteinassociated with repetitive sequences in the *T. brucei* genome. They claimed that this study has provided new information regarding the relevance of the repetitive region in the genome to chromosome integrity, telomere biology, chromosomal segregation and immune evasion strategies. These conclusions are based on high-quality research, and it is, basically, merits publication, provided that some major concerns, raised below, will be addressed before acceptance for publication.(1) The authors used TALE-YFP approach to examine the proteome associated with five different repetitive regions of the *T. brucei* genome and confirmed the binding of TALE-YFP with Chip-seq analyses. Ultimately, they got the list of proteins that bound to synthetic proteins, by affinity purification and LS-MS analysis and concluded that these proteins bind to different repetitive regions of the genome. There are two control proteins, one is TRF-YFP and the other KKT2-YFP, used to confirm the interactions. However, there are no experiment that confirms that the analysis gives some insight into the role of any putative or new protein in telomere biology, VSG gene regulation or chromosomal segregation. The proteins, which have already been reported by other studies, are mentioned. Although the author discovered many proteins in these repetitive regions, their role is yet unknown. It is recommended to take one or more of the new putative proteins from the repetitive elements and show whether or not they (1) bind directly to the specific repetitive sequence (e.g., by EMSA); (2) it is recommended that the authors will knockdown of one or a small sample of the new discovered proteins, which may shed light on their function at the repetitive region, as a proof of concept.

The main request from Referee 1 is for individual evaluation of protein-DNA interaction for a few candidates identified in our TALE-YFP affinity purifications, particularly using EMSA to identify binding to the DNA repeats used for the TALE selection. In our opinion, such an approach would not actually provide the validation anticipated by the reviewer. The power of TALE-YFP affinity selection is that it enriches for protein complexes that associate with the chromatin that coats the target DNA repetitive elements rather than only identifying individual proteins or components of a complex that directly bind to DNA assembled in chromatin.

The referee suggests we express recombinant proteins and perform EMSA for selected candidates, but many of the identified proteins are unlikely to directly bind to DNA – they are more likely to associate with a combination of features present in DNA and/or chromatin (e.g. specific histone variants or histone post-translational modifications). Of course, a positive result would provide some validation but only IF the tested protein can bind DNA in isolation – thus, a negative result would be uninformative.

In fact, our finding that KKT proteins are enriched using the 177R-TALE (minichromosome repeat sequence) identifies components of the trypanosome kinetochore known (KKT2) or predicted (KKT3) to directly bind DNA (Marciano et al., 2021; PMID: 34081090), and likewise the TelR-TALE identifies the TRF component that is known to directly associate with telomeric (TTAGGG)n repeats (Reis et al 2018; PMID: 29385523). This provides reassurance on the specificity of the selection, as does the lack of cross selectivity between different TALEs used (see later point 3 below). The enrichment of the respective DNA repeats quantitated in Figure 2B (originally Figure S1) also provides strong evidence for TALE selectivity.

It is very likely that most of the components enriched on the repetitive elements targeted by our TALE-YFP proteins do not bind repetitive DNA directly. The TRF telomere binding protein is an exception – but it is the only obvious DNA binding protein amongst the many proteins identified as being enriched in our TelR-TALE-YFP and TRF-YFP affinity selections.

The referee also suggests that follow up experiments using knockdown of the identified proteins found to be enriched on repetitive DNA elements would be informative. In our opinion, this manuscript presents the development of a new methodology previously not applied to trypanosomes, and referee 2 highlights the value of this methodological development which will be relevant for a large community of kinetoplastid researchers. In-depth follow-up analyses would be beyond the scope of this current study but of course will be pursued in future. To be meaningful such knockdown analyses would need to be comprehensive in terms of their phenotypic characterisation (e.g. quantitative effects on chromosome biology and cell cycle progression, rates and mechanism of recombination underlying antigenic variation, etc) – simple RNAi knockdowns would provide information on fitness but little more. This information is already publicly available from genome-wide RNAi screens (www.tritrypDB.org), with further information on protein location available from the genome-wide protein localisation resource (Tryptag.org). Hence basic information is available on all targets selected by the TALEs after RNAi knock down but in-depth follow-up functional analysis of several proteins would require specific targeted assays beyond the scope of this study.

(2) NonR-TALE-YFP does not have a binding site in the genome, but YFP protein should still be expressed by *T. brucei* clones with NLS. The authors have to explain why there is no signal detected in the nucleus, while a prominent signal was detected near kDNA (see Fig.2). Why is the expression of YFP in NonR-TALE almost not shown compared to other TALE clones?

The NonR-TALE-YFP immunolocalisation signal indeed is apparently located close to the kDNA and away from the nucleus. We are not sure why this is so, but the construct is sequence validated and correct. However, we note that artefactual localisation of proteins fused to a globular eGFP tag, compared to a short linear epitope V5 tag, near to the kinetoplast has been previously reported (Pyrih et al, 2023; PMID: 37669165).

The expression of NonR-TALE-YFP is shown in Supplementary Fig. S2 in comparison to other TALE proteins. Although it is evident that NonR-TALE-YFP is expressed at lower levels than other TALEs (the different TALEs have different expression levels), it is likely that in each case the TALE proteins would be in relative excess.

It is possible that the absence of a target sequence for the NonR-TALE-YFP in the nucleus affects its stability and cellular location. Understanding these differences is tangential to the aim of this study.

However, importantly, NonR-TALE-YFP is not the only control for used for specificity in our affinity purifications. Instead, the lack of cross-selection of the same proteins by different TALEs (e.g. TelR-TALE-YFP, 177R-TALE-YFP) and the lack of enrichment of any proteins of interest by the well expressed ingiR-TALE-YFP or 147R-TALE-YFP proteins each provide strong evidence for the specificity of the selection using TALEs, as does the enrichment of similar protein sets following affinity purification of the TelR-TALE-YFP and TRF-YFP proteins which both bind telomeric (TTAGGG)n repeats. Moreover, control affinity purifications to assess background were performed using cells that completely lack an expressed YFP protein which further support specificity (Figure 6).

We have added text to highlight these important points in the revised manuscript:

Page 8:

“However, the expression level of NonR-TALE-YFP was lower than other TALE-YFP proteins; this may relate to the lack of DNA binding sites for NonR-TALE-YFP in the nucleus.”

Page 8:

“NonR-TALE-YFP displayed a diffuse nuclear and cytoplasmic signal; unexpectedly the cytoplasmic signal appeared to be in the vicinity the kDNA of the kinetoplast (mitochrondria). We note that artefactual localisation of some proteins fused to an eGFP tag has previously been observed in *T. brucei* (Pyrih et al, 2023).”

Page 10:

Moreover, a similar set of enriched proteins was identified in TelR-TALE-YFP affinity purifications whether compared with cells expressing no YFP fusion protein (No-YFP), the NonR-TALE-YFP or the ingiR-TALE-YFP as controls (Fig. S7B, S8A; Tables S3, S4). Thus, the most enriched proteins are specific to TelR-TALE-YFP-associated chromatin rather than to the TALE-YFP synthetic protein module or other chromatin.

(3) As a proof of concept, the author showed that the TALE method determined the same interacting partners enrichment in TelR-TALE as compared to TRF-YFP. And they show the same interacting partners for other TALE proteins, whether compared with WT cells or with the NonR-TALE parasites. It may be because NonR-TALE parasites have almost no (or very little) YFP expression (see Fig. S3) as compared to other TALE clones and the TRF-YFP clone. To address this concern, there should be a control included, with proper YFP expression.

See response to point 2, but we reiterate that the ingi-TALE -YFP and 147R-TALE-YFP proteins are well expressed (western original Fig. S3 now Fig. S2) but few proteins are detected as being enriched or correspond to those enriched in TelR-TALE-YFP or TRF-YFP affinity purifications (see Fig. S9). Therefore, the ingi-TALE -YFP and 147R-TALE-YFP proteins provide good additional negative controls for specificity as requested. To further reassure the referee we have also included additional volcano plots which compare TelR-TALE-YFP, 70R-TALE-YFP or 177R-TALE-YFP to the ingiR-TALE-YFP affinity selection (new Figure S8). As with No-YFP or NonR-TALE-YFP controls, the use of ingiR-TALE-YFP as a negative control demonstrates that known telomere associated proteins are enriched in TelR-TALE-YFP affinity purification, RPA subunits enriched with 70R-TALE-YFP and Kinetochore KKT poroteins enriched with 177RTALE-YFP. These analyses demonstrate specificity in the proteins enriched following affinity purification of our different TALE-YFPs and provide support to strengthen our original findings.

We now refer to use of No-YFP, NonR-TALE-YFP, and ingiR-TALE -YFP as controls for comparison to TelR-TALE-YFP, 70R-TALE-YFP or 177R-TALE-YFP in several places:

Page10:

“Moreover, a similar set of enriched proteins was identified in TelR-TALE-YFP affinity purifications whether compared with cells expressing no YFP fusion protein (No-YFP), the NonR-TALE-YFP or the ingiR-TALE-YFP as controls (Fig. S7B, S8A; Tables S3, S4).”

Page 11:

“Thus, the nuclear ingiR-TALE-YFP provides an additional chromatin-associated negative control for affinity purifications with the TelR-TALE-YFP, 70R-TALE-YFP and 177R-TALE-YFP proteins (Fig. S8).”

“Proteins identified as being enriched with 70R-TALE-YFP (Figure 6D) were similar in comparisons with either the No-YFP, NonR-TALE-YFP or ingiR-TALE-YFP as negative controls.”

Top Page 12:

“The same kinetochore proteins were enriched regardless of whether the 177R-TALE proteomics data was compared with No-YFP, NonR-TALE or ingiR-TALE-YFP controls.”

Discussion Page 13:

“Regardless, the 147R-TALE and ingiR-TALE proteins were well expressed in *T. brucei* cells, but their affinity selection did not significantly enrich for any relevant proteins. Thus, 147R-TALE and ingiR-TALE provide reassurance for the overall specificity for proteins enriched TelR-TALE, 70R-TALE and 177R-TALE affinity purifications.”

(4) After the artificial expression of repetitive sequence binding five-TALE proteins, the question is if there is any competition for the TALE proteins with the corresponding endogenous proteins? Is there any effect on parasite survival or health, compared to the control after the expression of these five TALEs YFP protein? It is recommended to add parasite growth curves, for all the TALE proteins expressing cultures.

Growth curves for cells expressing TelR-TALE-YFP, 177R-TALE-YFP and ingiR-TALE-YFP are now included (New Fig S3A). No deficit in growth was evident while passaging 70R-TALE-YFP, 147R-TALE-YFP, NonR-TALE-YFP cell lines (indeed they grew slightly better than controls).

The following text has been added page 8:

“Cell lines expressing representative TALE-YFP proteins displayed no fitness deficit (Fig. S3A).”

(5) Since the experiments were performed using whole-cell extracts without prior nuclear fractionation, the authors should consider the possibility that some identified proteins may have originated from compartments other than the nucleus. Specifically, the detection of certain binding proteins might reflect sequence homology (or partial homology) between mitochondrial DNA (maxicircles and minicircles) and repetitive regions in the nuclear genome. Additionally, the lack of subcellular separation raises the concern that cytoplasmic proteins could have been co-purified due to whole cell lysis, making it challenging to discern whether the observed proteome truly represents the nuclear interactome.

In our experimental design, we confirmed bioinformatically that the repeat sequences targeted were not represented elsewhere in the nuclear or mitochondrial genome (kDNA). The absence of subcellular fractionation could result in some cytoplasmic protein selection, but this is unlikely since each TALE targets a specific DNA sequence but is otherwise identical such that cross-selection of the same contaminating protein set would be anticipated if there was significant non-specific binding. We have previously successfully affinity selected 15 chromatin modifiers and identified associated proteins without major issues concerning cytoplasmic protein contamination (Staneva et al 2021 and 2022; PMID: 34407985 and 36169304). Of course, the possibility that some proteins are contaminants will need to be borne in mind in any future follow-up analysis of proteins of interest that we identified as being enriched on specific types of repetitive element in *T. brucei*. Proteins that are also detected in negative control, or negative affinity selections such as No-YFP, NoR-YFP, IngiR-TALE or 147R-TALE must be disregarded.

(6) Should the authors qualify some of their claims as preliminary or speculative, or remove them altogether?As mentioned earlier, the author claimed that this study has provided new information concerning telomere biology, chromosomal segregation mechanisms, and immune evasion strategies. But there are no experiments that provides a role for any unknown or known protein in these processes. Thus, it is suggested to select one or two proteins of choice from the list and validate their direct binding to repetitive region(s), and their role in that region of interaction.

As highlighted in response to point 1 the suggested validation and follow up experiments may well not be informative and are beyond the scope of the methodological development presented in this manuscript. Referee 2 describes the study in its current form as “a significant conceptual and technical advancement” and “This approach enhances our understanding of chromatin organization in these regions and provides a foundation for investigating the functional roles of associated proteins in parasite biology.”

The Referee’s phrase ‘validate their direct binding to repetitive region(s)’ here may also mean to test if any of the additional proteins that we identified as being enriched with a specific TALE protein actually display enrichment over the repeat regions when examined by an orthogonal method. A key unexpected finding was that kinetochore proteins including KKT2 are enriched in our affinity purifications of the 177R-TALE-YFP that targets 177bp repeats (Figure 6F). By conducting ChIP-seq for the kinetochore specific protein KKT2 using YFP-KKT2 we confirmed that KKT2 is indeed enriched on 177bp repeat DNA but not flanking DNA (Figure 7). Moreover, several known telomere-associated proteins are detected in our affinity selections of TelRTALE-YFP (Figure 6B, FigS6; see also Reis et al, 2018 Nuc. Acids Res. PMID: 29385523; Weisert et al, 2024 Sci. Reports PMID: 39681615).

Would additional experiments be essential to support the claims of the paper? Request additional experiments only where necessary for the paper as it is, and do not ask authors to open new lines of experimentation.The answer for this question depends on what the authors want to present as the achievements of the present study. If the achievement of the paper was is the creation of a new tool for discovering new proteins, associated with the repeat regions, I recommend that they add a proof for direct interactions between a sample the newly discovered proteins and the relevant repeats, as a proof of concept discussed above, However, if the authors like to claim that the study achieved new functional insights for these interactions they will have to expand the study, as mentioned above, to support the proof of concept.

See our response to point 1 and the point we labelled ‘6’ above.

Are the suggested experiments realistic in terms of time and resources? It would help if you could add an estimated cost and time investment for substantial experiments.I think that they are realistic. If the authors decided to check the capacity of a small sample of proteins (which was unknown before as a repetitive region binding proteins) to interacts directly with the repeated sequence, it will substantially add of the study (e.g., by EMSA; estimated time: 1 months). If the authors will decide to check the also the function of one of at least one such a newly detected proteins (e.g., by KD), I estimate the will take 3-6 months.

As highlighted previously the proposed EMSA experiment may well be uninformative for protein complex components identified in our study or for isolated proteins that directly bind DNA in the context of a complex and chromatin. RNAi knockdown data and cell location data (as well as developmental expression and orthology data) is already available through tritrypDB.org and trtyptag.org

Are the data and the methods presented in such a way that they can be reproduced? YesAre the experiments adequately replicated, and statistical analysis adequate?The authors did not mention replicates. There is no statistical analysis mentioned.

The figure legends indicate that all volcano plots of TALE affinity selections were derived from three biological replicates. Cutoffs used for significance: *P* < 0.05 (Student's *t*-test).

For ChiP-seq two biological replicates were analysed for each cell line expressing the specific YFP tagged protein of interest (TALE or KKT2). This is now stated in the relevant figure legends – apologies for this oversight. The resulting data are available for scrutiny at GEO: GSE295698.

Minor comments:Specific experimental issues that are easily addressable.The following suggestions can be incorporated:(1) Page 18, in the material method section author mentioned four drugs: Blasticidine, Phleomycin and G418, and hygromycin. It is recommended to mention the purpose of using these selective drugs for the parasite. If clonal selection has been done, then it should also be mentioned.

We erroneously added information on several drugs used for selection in our labaoratory. In fact all TALE-YFP construct carry the Bleomycin resistance genes which we select for using Phleomycin. Also, clones were derived by limiting dilution immediately after transfection. We have amended the text accordingly:

Page 17/18:

“Cell cultures were maintained below 3 x 106 cells/ml. Pleomycin 2.5 µg/ml was used to select transformants containing the TALE construct BleoR gene.”

“Electroporated bloodstream cells were added to 30 ml HMI-9 medium and two 10-fold serial dilutions were performed in order to isolate clonal Pleomycin resistant populations from the transfection. 1 ml of transfected cells were plated per well on 24-well plates (1 plate per serial dilution) and incubated at 37°C and 5% CO2 for a minimum of 6 h before adding 1 ml media containing 2X concentration Pleomycin (5 µg/ml) per well.”

(2) In the method section the authors mentioned that there is only one site for binding of NonR-TALE in the parasite genome. But in Fig. 1C, the authors showed zero binding site. So, there is one binding site for NonR-TALE-YFP in the genome or zero?

We thank the reviewer for pointing out this discrepancy. We have checked the latest Tb427v12 genome assembly for predicted NonR-TALE binding sites and there are no exact matches. We have corrected the text accordingly.

Page 7:

“A control NonR-TALE protein was also designed which was predicted to have no target sequence in the *T. brucei* genome.”

Page 17:

“A control NonR-TALE predicted to have no recognised target in the *T. brucei* geneome was designed as follows: BLAST searches were used to identify exact matches in the TREU927 reference genome. Candidate sequences with one or more match were discarded.”

(3) The authors used two different anti-GFP antibodies, one from Roche and the other from Thermo Fisher. Why were two different antibodies used for the same protein?

We have found that only some anti-GFP antibodies are effective for affinity selection of associated proteins, whereas others are better suited for immunolocalisation. The respective suppliers’ antibodies were optimised for each application.

(4) Page 6: in the introduction, the authors give the number of total VSG genes as 2,634. Is it known how many of them are pseudogenes?

This value corresponds to the number reported by Consentino et al. 2021 (PMID: 34541528) for subtelomeric VSGs, which is similar to the value reported by Muller et al 2018 (PMID: 30333624) (2486), both in the same strain of trypanosomes as used by us. Based on the earlier analysis by Cross et al (PMID: 24992042), 80% of the identified VSGs in their study (2584) are pseudogenes. This approximates to the estimation by Consentino of 346/2634 (13%) being fully functional VSG genes at subtelomeres, or 17% when considering VSGs at all genomic locations (433/2872).

(5) I found several typos throughout the manuscript.

Thank you for raising this, we have read through the manuscipt several times and hopefully corrected all outstanding typos.

(6) Fig. 1C: Table: below TOTAL 2nd line: the number should be 1838 (rather than 1828)

Corrected- thank you.

- Are prior studies referenced appropriately? Yes- Are the text and figures clear and accurate? Yes- Do you have suggestions that would help the authors improve the presentation of their data and conclusions? Suggested above

**Reviewer #1 (Significance):**

Describe the nature and significance of the advance (e.g., conceptual, technical, clinical) for the field:This study represents a significant conceptual and technical advancement by employing a synthetic TALE DNA-binding protein tagged with YFP to selectively identify proteins associated with five distinct repetitive regions of *T. brucei* chromatin. To the best of my knowledge, it is the first report to utilize TALE-YFP for affinity-based isolation of protein complexes bound to repetitive genomic sequences in *T. brucei*. This approach enhances our understanding of chromatin organization in these regions and provides a foundation for investigating the functional roles of associated proteins in parasite biology. Importantly, any essential or unique interacting partners identified could serve as potential targets for therapeutic intervention.- Place the work in the context of the existing literature (provide references, where appropriate). I agree with the information that has already described in the submitted manuscript, regarding its potential addition of the data resulted and the technology established to the study of VSGs expression, kinetochore mechanism and telomere biology.- State what audience might be interested in and influenced by the reported findings. These findings will be of particular interest to researchers studying the molecular biology of kinetoplastid parasites and other unicellular organisms, as well as scientists investigating chromatin structure and the functional roles of repetitive genomic elements in higher eukaryotes.- (1) Define your field of expertise with a few keywords to help the authors contextualize your point of view. Protein-DNA interactions/ chromatin/ DNA replication/ Trypanosomes- (2) Indicate if there are any parts of the paper that you do not have sufficient expertise to evaluate. None
**Reviewer #2 (Evidence, reproducibility and clarity):**
SummaryCarloni et al. comprehensively analyze which proteins bind repetitive genomic elements in *Trypanosoma brucei*. For this, they perform mass spectrometry on custom-designed, tagged programmable DNA-binding proteins. After extensively verifying their programmable DNA-binding proteins (using bioinformatic analysis to infer target sites, microscopy to measure localization, ChIP-seq to identify binding sites), they present, among others, two major findings: (1) 14 of the 25 known *T. brucei* kinetochore proteins are enriched at 177bp repeats. As *T. brucei's* 177bp repeatcontaining intermediate-sized and mini-chromosomes lack centromere repeats but are stable over mitosis, Carloni et al. use their data to hypothesize that a 'rudimentary' kinetochore assembles at the 177bp repeats of these chromosomes to segregate them. (2) 70bp repeats are enriched with the Replication Protein A complex, which, notably, is required for homologous recombination. Homologous recombination is the pathway used for recombination-based antigenic variation of the 70bp-repeat-adjacent variant surface glycoproteins.Major CommentsNone. The experiments are well-controlled, claims well-supported, and methods clearly described. Conclusions are convincing.

Thank you for these positive comments.

Minor Comments(1) Fig. 2 - I couldn't find an uncropped version showing multiple cells. If it exists, it should be linked in the legend or main text; Otherwise, this should be added to the supplement.

The images presented represent reproducible analyses, and independently verified by two of the authors. Although wider field of view images do not provide the resolution to be informative on cell location, as requested we have provided uncropped images in new Fig. S4 for all the cell lines shown in Figure 2A.

In addition, we have included as supplementary images (Fig. S3B) additional images of TelRTALE-YFP, 177R-TALE-YFP and ingiR-TALE YFP localisation to provide additional support their observed locations presented in Figure 1. The set of cells and images presented in Figure 2A and in Fig S3B were prepared and obtained by a different authors, independently and reproducibly validating the location of the tagged protein.

(2) I think Suppl. Fig. 1 is very valuable, as it is a quantification and summary of the ChIP-seq data. I think the authors could consider making this a panel of a main figure. For the main figure, I think the plot could be trimmed down to only show the background and the relevant repeat for each TALE protein, leaving out the non-target repeats. (This relates to minor comment 6.) Also, I believe, it was not explained how background enrichment was calculated.

We are grateful for the reviewer’s positive view of original Fig. S1 and appreciate the suggestion. We have now moved these analysis to part B of main Figure 2 in the revised manuscript – now Figure 2B. We have also provided additional details in the Methods section on the approaches used to assess background enrichment.

Page 19:

“Background enrichment calculation

The genome was divided into 50 bp sliding windows, and each window was annotated based on overlapping genomic features, including CIR147, 177 bp repeats, 70 bp repeats, and telomeric (TTAGGG)n repeats. Windows that did not overlap with any of these annotated repeat elements were defined as "background" regions and used to establish the baseline ChIP-seq signal. Enrichment for each window was calculated using bamCompare, as log₂(IP/Input). To adjust for background signal amongst all samples, enrichment values for each sample were further normalized against the corresponding No-YFP ChIP-seq dataset.”

Note: While revising the manuscript we also noticed that the script had a nomalization error. We have therefore included a corrected version of these analyses as Figure 2B (old Fig. S1)

(3) Generally, I would plot enrichment on a log2 axis. This concerns several figures with ChIP-seq data.

Our ChIP-seq enrichment is calculated by bamCompare. The resulting enrichment values are indeed log2 (IP/Input). We have made this clear in the updated figures/legends.

(4) Fig. 4C - The violin plots are very hard to interpret, as the plots are very narrow compared to the line thickness, making it hard to judge the actual volume. For example, in Centromere 5, YFP-KKT2 is less enriched than 147R-TALE over most of the centromere with some peaks of much higher enrichment (as visible in panel B), however, in panel C, it is very hard to see this same information. I'm sure there is some way to present this better, either using a different type of plot or by improving the spacing of the existing plot.

We thank the reviewer for this suggestion; we have elected to provide a Split-Violin plot instead. This improves the presentation of the data for each centromere. The original violin plot in Figure 4C has been replaced with this Split-Violin plot (still Figure 4C).

(5) Fig. 6 - The panels are missing an x-axis label (although it is obvious from the plot what is displayed).Maybe the "WT NO-YFP vs" part that is repeated in all the plot titles could be removed from the title and only be part of the x-axis label?

In fact, to save space the X axis was labelled inside each volcano plot but we neglected to indicate that values are a log2 scale indicating enrichment. This has been rectified – see Figure 6, and Fig. S7, S8 and S9.

(6) Fig. 7 - I would like to have a quantification for the examples shown here. In fact, such a quantification already exists in Suppl. Figure 1. I think the relevant plots of that quantification (YFPKKT2 over 177bp-repeats and centromere-repeats) with some control could be included in Fig. 7 as panel C. This opportunity could be used to show enrichment separated out for intermediate-sized, mini-, and megabase-chromosomes. (relates to minor comment 2 & 8)

The CIR147 sequence is found exclusively on megabase-sized chromosomes, while the 177 bp repeats are located on intermediate- and mini-sized chromosomes. Due to limitations in the current genome assembly, it is not possible to reliably classify all chromosomes into intermediate- or mini- sized categories based on their length. Therefore, original Supplementary Fig. S1 presented the YFP-KKT2 enrichment over CIR147 and 177 bp repeats as a representative comparison between megabase chromosomes and the remaining chromosomes (corrected version now presented as main Figure 2B). Additionally, to allow direct comparison of YFP-KKT2 enrichment on CIR147 and 177 bp repeats we have included a new plot in Figure 7C which shows the relative enrichment of YFP-KKT2 on these two repeat types.

We have added the following text , page 12:

“Taking into account the relative to the number of CIR147 and 177 bp repeats in the current *T. brucei* genome (Cosentino et al., 2021; Rabuffo et al., 2024), comparative analyses demonstrated that YFP-KKT2 is enriched on both CIR147 and 177 bp repeats (Figure 7C).”

(7) Suppl. Fig. 8 A - I believe there is a mistake here: KKT5 occurs twice in the plot, the one in the overlap region should be KKT1-4 instead, correct?

Thanks for spotting this. It has been corrected

(8) The way that the authors mapped ChIP-seq data is potentially problematic when analyzing the same repeat type in different regions of the genome. The authors assigned reads that had multiple equally good mapping positions to one of these mapping positions, randomly.This is perfectly fine when analysing repeats by their type, independent of their position on the genome, which is what the authors did for the main conclusions of the work.However, several figures show the same type of repeat at different positions in the genome. Here, the authors risk that enrichment in one region of the genome 'spills' over to all other regions with the same sequence. Particularly, where they show YFP-KKT2 enrichment over intermediate- and mini-chromosomes (Fig. 7) due to the spillover, one cannot be sure to have found KKT2 in both regions.Instead, the authors could analyze only uniquely mapping reads / read-pairs where at least one mate is uniquely mapping. I realize that with this strict filtering, data will be much more sparse. Hence, I would suggest keeping the original plots and adding one more quantification where the enrichment over the whole region (e.g., all 177bp repeats on intermediate-/mini-chromosomes) is plotted using the unique reads (this could even be supplementary). This also applies to Fig. 4 B & C.

We thank the reviewer for their thoughtful comments. Repetitive sequences are indeed challenging to analyze accurately, particularly in the context of short read ChIP-seq data. In our study, we aimed to address YFP-KKT2 enrichment not only over CIR147 repeats but also on 177 bp repeats, using both ChIP-seq and proteomics using synthetic TALE proteins targeted to the different repeat types. We appreciate the referees suggestion to consider uniquely mapped reads, however, in the updated genome assembly, the 177 bp repeats are frequently immediately followed by long stretches of 70 bp repeats which can span several kilobases. The size and repetitive nature of these regions exceeds the resolution limits of ChIP-seq. It is therefore difficult to precisely quantify enrichment across all chromosomes.

Additionally, the repeat sequences are highly similar, and relying solely on uniquely mapped reads would result in the exclusion of most reads originating from these regions, significantly underestimating the relative signals. To address this, we used Bowtie2 with settings that allow multi-mapping, assigning reads randomly among equivalent mapping positions, but ensuring each read is counted only once. This approach is designed to evenly distribute signal across all repetitive regions and preserve a meaningful average.

Single molecule methods such as DiMeLo (Altemose et al. 2022; PMID: 35396487) will need to be developed for *T. brucei* to allow more accurate and chromosome specific mapping of kinetochore or telomere protein occupancy at repeat-unique sequence boundaries on individual chromosomes.

**Reviewer #2 (Significance):**
This work is of high significance for chromosome/centromere biology, parasitology, and the study of antigenic variation. For chromosome/centromere biology, the conceptual advancement of different types of kinetochores for different chromosomes is a novelty, as far as I know. It would certainly be interesting to apply this study as a technical blueprint for other organisms with minichromosomes or chromosomes without known centromeric repeats. I can imagine a broad range of labs studying other organisms with comparable chromosomes to take note of and build on this study. For parasitology and the study of antigenic variation, it is crucial to know how intermediate- and mini-chromosomes are stable through cell division, as these chromosomes harbor a large portion of the antigenic repertoire. Moreover, this study also found a novel link between the homologous repair pathway and variant surface glycoproteins, via the 70bp repeats. How and at which stages during the process, 70bp repeats are involved in antigenic variation is an unresolved, and very actively studied, question in the field. Of course, apart from the basic biological research audience, insights into antigenic variation always have the potential for clinical implications, as *T. brucei* causes sleeping sickness in humans and nagana in cattle. Due to antigenic variation, *T. brucei* infections can be chronic.

Thank you for supporting the novelty and broad interest of our manuscript

My field of expertise / Point of view:I'm a computer scientist by training and am now a postdoctoral bioinformatician in a molecular parasitology laboratory. The laboratory is working on antigenic variation in *T. brucei*. The focus of my work is on analyzing sequencing data (such as ChIP-seq data) and algorithmically improving bioinformatic tools.